# Corruption-Tolerant Asynchronous Q-Learning with Near-Optimal Rates

**Sreejeet Maity**
Electrical and Computer Engineering
North Carolina State University
Raleigh, NC 27606
`smaity2@ncsu.edu`

**Aritra Mitra**
Electrical and Computer Engineering
North Carolina State University
Raleigh, NC 27606
`amitra2@ncsu.edu`

## Abstract

We study the problem of learning the optimal policy in a discounted, infinite-horizon reinforcement learning (RL) setting in the presence of adversarially corrupted rewards. To address this problem, we develop a novel robust variant of the Q-learning algorithm and analyze it under the challenging asynchronous sampling model with time-correlated data. Despite corruption, we prove that the finite-time guarantees of our approach match existing bounds, up to an additive term that scales with the fraction of corrupted samples. We also establish an information-theoretic lower bound, revealing that our guarantees are near-optimal. Notably, our algorithm is agnostic to the underlying reward distribution and provides the first finite-time robustness guarantees for asynchronous Q-learning. A key element of our analysis is a refined Azuma-Hoeffding inequality for almost-martingales, which may have broader applicability in the study of RL algorithms.

## 1 Introduction

In a typical reinforcement learning (RL) problem, a learning agent interacts sequentially with an environment modeled as a Markov Decision Process (MDP). Each interaction involves the agent playing an action and receiving feedback in the form of a reward for the action taken. Using such feedback, the agent gains a better understanding of the quality of the actions, allowing it to eventually learn an optimal decision-making policy. The formalism described above finds use in a variety of practical applications, spanning finance, medicine, recommendation systems, autonomous driving, robotics, and most recently, training large language models using human feedback. In each of these applications, *the effectiveness of the learned policy depends crucially on the quality of the feedback data (rewards)* used to train the policy. In real-world applications, however, data can be noisy and can contain outliers: human feedback can be biased and have malicious intent, recommendation systems can be skewed by fake users, and sensor data in an autonomous vehicle can be prone to measurement errors and be corrupted by an adversary. If precautions are not taken to contend with "bad data", then the consequences can be dire, especially for safety-critical applications. Motivated by this concern, we revisit the classical RL problem from the perspective of *adversarial robustness* and study a scenario where a portion of the rewards observed by the learner can be corrupted *arbitrarily*. For this scenario, we wish to understand to what extent one can hope to still learn a (near)-optimal policy. Surprisingly, despite the popularity of the RL paradigm, a complete theoretical understanding of this question seems to be lacking in the current literature, especially for the scenario where data are collected in an online, sequential manner. Our work in this paper contributes to filling this gap.

We consider an infinite-horizon discounted RL problem, where an agent collects data from the environment based on a behavior/sampling policy, as is done with popular RL algorithms such as Q-learning [1]. We depart from the standard RL observation model by allowing the rewards to be

corrupted based on a fixed corruption probability $\varepsilon \in [0, 1/2)$: at each time-step, with probability $1 - \varepsilon$, the learner (agent) observes a reward sampled from the true reward distribution associated with the current state and action, and with probability $\varepsilon$, it observes a sample from an arbitrary adversarial distribution. Importantly, we put no restrictions at all on the adversarial distribution, allowing for potentially unbounded attack signals. Furthermore, we allow the true reward distributions to be *heavy-tailed*, requiring them to admit no more than a finite second moment. It should be noted here that our way of modeling corruption is inspired directly by the Huber model from robust statistics [2, 3]. Furthermore, similar corruption models have been extensively studied for the simpler bandits setting [4–9], and more recently in offline RL with human feedback [10]. However, when it comes to learning an optimal policy in the infinite-horizon discounted setting we consider here with online, sequential data, the effect of such an attack model remains completely unexplored. Since an optimal policy can be extracted by learning the optimal state-action value function [11], we ask two concrete questions: Subject to our corruption model: (i) *Can one still reliably estimate the optimal state-action value function?* (ii) *What is a fundamental lower bound on estimation accuracy in this setting?* Our contributions described below comprehensively address these questions.

- **Novel Robust Q-Learning Algorithm.** In Section 3, we start by considering a setting where bounds on the first and second moments of the true reward distributions are known to the learner. For this setting, we propose a new algorithm called `Robust Async-Q` that comprises two main ingredients. The first idea is to leverage the recent univariate trimmed mean estimator from [12] to maintain running estimates of the mean rewards for each state-action pair of the MDP, using historical data for such pairs. However, this idea is not enough on its own since the guarantees associated with robust mean estimation are probabilistic in nature, and, as such, may not hold on rare, extreme events. To control the errors introduced by adversarial contamination on such rare events, we employ a second layer of safety that involves keeping track of "typical" regions that contain the reward mean estimates; estimates that fall outside the typical regions are rejected. The size of these typical regions - as captured by an *adaptive threshold* - shrinks as the learner acquires more samples.

For the case where bounds on the reward statistics are *unknown* a priori, constructing the adaptive threshold accurately becomes much trickier. In Section 4, we propose a simple modification to `Robust Async-Q` that addresses this challenge by using a "slowly growing" function of time as a proxy for such bounds. *Overall, we prescribe a framework for constructing robust empirical estimates of the Bellman optimality operator using noisy, corrupted data collected online.*

- **Finite-Time Rates under I.I.D. Sampling.** To build intuition, we start by analyzing `Robust Async-Q` under a simplified i.i.d. sampling model, commonly used in previous RL works [13–16]. In Theorems 2 and 4, we provide high-probability finite time rates for `Robust Async-Q` with known and unknown reward statistics, respectively. Given $T$ samples, in each case, our bounds match the known optimal rate [17–19] of $\tilde{O}(1/\sqrt{T})$, up to a small additive term on the order of $O(\sqrt{\varepsilon})$, where $\varepsilon$ is the probability of corruption. Interestingly, our bounds also reveal how the effect of asynchronous sampling can inflate the corruption-induced term. *To our knowledge, Theorems 2 and 4 provide the first formal guarantees of adversarial robustness for asynchronous Q-learning.*

- **Fundamental Lower Bound.** One might ask whether the $O(\sqrt{\varepsilon})$ term in our upper-bound is unavoidable. In Theorem 3, we settle this question by providing an information-theoretic fundamental lower bound, revealing that an $\Omega(\sqrt{\varepsilon})$ error in the estimation of the optimal state-action value function is *unavoidable*. **Collectively, our results are significant in that they reveal that `Robust Async-Q` achieves near-optimal finite-time guarantees for Q-learning under adversarial corruption.**

- **Finite-Time Rates under Markov Sampling.** In Section 4.1, we study our setting in full generality by considering the challenging single-trajectory Markovian sampling model with time-correlated data. In Theorem 5, we prove that one can nearly recover the same bounds as in the i.i.d. setting, up to an inflation in the $\tilde{O}(1/\sqrt{T})$ term caused by the mixing time of the underlying Markov chain; notably this inflation is consistent with prior bounds in the absence of corruption [18].

- **Novel Proof Techniques.** Arriving at our results involves several new proof ingredients. Even with i.i.d. sampling and known reward statistics, some work is needed to account for the fact that under the asynchronous sampling model, the number of times each state-action pair has been sampled (up to a given time-step) is a *random variable*, precluding the direct use of robust mean estimation bounds. To overcome this issue, we use Bernstein's inequality to control the number of visits to each state-action pair. A key new step in our analysis is to argue that after a certain burn-in time, no estimates will be rejected (due to thresholding) on a good event of sufficient measure. When

the reward statistics are unknown a priori, the use of slowly growing functions of time as their proxies introduces significant new challenges. In particular, as we explain in Section 4, using the standard version of the Azuma-Hoeffding inequality - which is what is done in existing Q-learning analyses [18] - will unfortunately lead to vacuous bounds in our setting. Furthermore, relatively well-known variants of the Azuma-Hoeffding inequality for discrete probability spaces [20], and sub-Gaussian martingale differences [21] also prove to be inadequate for our purposes. To overcome this challenge, we show how a refined variant of the Azuma-Hoeffding inequality from [22] can be carefully exploited to preserve near-optimal bounds; *we are unaware of the use of this new tool in any prior RL work*, and believe that it might be more broadly applicable. Finally, to handle the challenging single-trajectory Markovian data setting, we combine the aforementioned ideas with a coupling technique that is inspired by recent work [23, 24].

**Summary.** To sum up, we provide the first principled and comprehensive study of adversarial robustness in RL for the infinite-horizon, discounted setting with asynchronous Markovian data. Our new algorithms and analysis techniques, complemented by nearly matching upper and lower-bounds, paint a fairly complete picture for this setting.

**Related Work.** We now discuss the most relevant works on corruption-robust RL here, and relegate a more detailed survey to Appendix A. The topic of reward corruption has been explored in several papers on bandits [4–7, 25–27, 8, 28, 9]. In the context of MDPs, data corruption in online, finite-horizon episodic RL problems is studied in [29–32], where performance is measured by cumulative regret and the algorithms are variants of either Upper-Confidence-Based (UCB) or Action-Elimination strategies. The infinite-horizon discounted setting we study here *differs fundamentally* in terms of the notion of performance (sample-complexity), and also in terms of the algorithm design principle, which is rooted in stochastic approximation theory. Corruption-robust algorithms in the offline setting or with access to a generative model/simulator are considered in [33, 34, 10, 35], where batched data tuples are collected offline in an i.i.d. manner. In sharp contrast, we need to contend with a much more challenging observation model, where *heavy-tailed and corrupted* data arrives in an online, sequential manner as part of a *single trajectory*, and the state-action pairs are visited asynchronously, creating the problem of *partial observability*. Finally, we note that the issue of handling just heavy-tailed rewards (without adversarial corruption) has been studied in problem settings different from ours: for offline RL in [36], for episodic RL in [37], and for policy evaluation in [38].

## 2 Background and Problem Formulation

We start by providing the basic background on RL, and then proceed to describe our problem of interest. We consider a $\gamma$-discounted infinite-horizon Markov Decision Process (MDP) $\mathcal{M} = (\mathcal{S}, \mathcal{A}, \mathcal{P}, R, \gamma)$, where $\mathcal{S}$ is a finite state space, $\mathcal{A}$ is a finite action space, $\mathcal{P}$ is a set of state transition kernels, $R$ is a reward function, and $\gamma \in (0, 1)$ is a discount factor. When in state $s \in \mathcal{S}$ the learner plays an action $a \in \mathcal{A}$, it observes a new state $s'$ drawn from $\mathcal{P}(\cdot|s, a)$, and a stochastic reward sample $r(s, a)$ drawn from a reward distribution $\mathcal{R}(s, a)$. The noisy reward $r(s, a)$ is unbiased with mean equal to the true expected reward $R(s, a)$ for state-action pair $(s, a)$, and variance $\sigma^2(s, a)$, i.e., $\mathbb{E}[r(s, a)] = R(s, a)$, and $\mathbb{E}[(r(s, a) - R(s, a))^2] = \sigma^2(s, a)$. We assume that the mean rewards and variances are uniformly bounded, i.e., there exist $\bar{R}, \bar{\sigma} \geq 1$ such that $|R(s, a)| \leq \bar{R}$ and $\sigma^2(s, a) \leq \bar{\sigma}^2, \forall(s, a) \in \mathcal{S} \times \mathcal{A}$. A policy $\mu : \mathcal{S} \to \Delta(\mathcal{A})$ is a mapping from the states to a space of probability distributions over actions, denoted by $\Delta(\mathcal{A})$. The quality of a policy $\mu$ is captured by an expected discounted infinite-horizon cumulative reward known as the value function $V^\mu$, defined as

$$V^\mu(s) = \mathbb{E}\left[\sum_{t=0}^\infty \gamma^t R(s_t, a_t) \,\middle|\, s_0 = s, \mu\right], \tag{1}$$

where $s_t$ and $a_t$ are the state and action at time $t$, respectively, under the action of the policy $\mu$ on the MDP $\mathcal{M}$. The goal of the learner is to find a policy $\mu$ that maximizes the value function $V^\mu$ for all states, *without knowledge* of the transition kernels $\mathcal{P}$ and reward functions $R$ of the underlying MDP. To explain how this is done, we will need to introduce the notion of a state-action value function $Q^\mu$ for a policy $\mu$, defined as $Q^\mu(s, a) = \mathbb{E}\left[\sum_{t=0}^\infty \gamma^t R(s_t, a_t) \,\middle|\, (s_0, a_0) = (s, a), \mu\right]$. The celebrated $Q$-learning algorithm [1] uses data collected by a suitable behavior/sampling policy $\mu$ to iteratively maintain an estimate of the optimal state-action value function, denoted by $Q^*$. It turns out that $Q^*$ is the fixed point of a contractive operator known as the Bellman optimality operator [11]. Using this

contraction property, classical asymptotic results [39, 40] established that the sequence of iterates generated by Q-learning converges to $Q^*$ almost surely (under suitable assumptions on $\mu$). More recently, finite-time rates have been established [17–19], revealing that when run for $T$ iterations, the final iterate of Q-learning converges to $Q^*$ at a rate of $\tilde{O}(1/\sqrt{T})$, with high probability. Once $Q^*$ is known, an optimal policy can be determined by playing actions greedily with respect to $Q^*$ [41].

**Adversarially Corrupted Reward Model.** Our formulation departs from the standard setting described above in two main ways. First, classical results on Q-learning either assume deterministic rewards or "light-tailed" noisy rewards with sub-Gaussian reward distributions. In contrast, our formulation requires the reward distributions $\mathcal{R}(s, a)$ to admit only up to a finite second moment, and nothing more. Thus, *the true reward distributions are allowed to be heavy-tailed.* More importantly, we allow a portion of the reward data to be corrupted *arbitrarily* by an adversary. To explain the corruption model precisely, suppose that data are collected based on a stochastic behavior policy $\mu$, such that $\mu(a|s) > 0, \forall s \in \mathcal{S}, \forall a \in \mathcal{A}$. Upon interacting with the MDP $\mathcal{M}$, the policy $\mu$ induces a Markov chain. Let $s_t$ be the state of this Markov chain at time $t$. Then, in the standard Q-learning setting, at each time-step $t$, the learner observes the data tuple $(s_t, a_t, s_{t+1})$, and noisy reward $r_t(s_t, a_t)$, where $a_t \sim \mu(\cdot|s_t)$, $s_{t+1} \sim \mathcal{P}(\cdot|s_t, a_t)$, and $r_t(s_t, a_t) \sim \mathcal{R}(s_t, a_t)$. Here, we assume that the noise process $\{R(s_t, a_t) - r_t(s_t, a_t)\}$ is independent over time and of all other sources of randomness. In our setting, the learner still observes $(s_t, a_t, s_{t+1})$, but now receives a Huber-contaminated reward $y_t(s_t, a_t)$ generated as follows. At time $t$, a biased coin with probability of heads $1 - \varepsilon$ is tossed independently of the past, and all other sources of randomness in the problem; here $\varepsilon \in [0, 1/2)$ is a fixed probability that captures the fraction of corrupted samples. If the coin lands heads, $y_t(s_t, a_t)$ is drawn from the true reward distribution $\mathcal{R}(s_t, a_t)$. If it lands tails, $y_t(s_t, a_t)$ is drawn from an *unconstrained and arbitrary* adversarial distribution $\mathcal{Q}$ that can depend on history, and be time and state-action pair dependent. In other words, if $y_t(s_t, a_t)$ is drawn from $\mathcal{Q}$, it can be arbitrary (and hence, potentially unbounded). Concretely, we write $y_t(s_t, a_t) \sim (1-\varepsilon)\mathcal{R}(s_t, a_t) + \varepsilon\mathcal{Q}$, where the notation $(1 - \varepsilon)\mathcal{P}_1 + \varepsilon\mathcal{P}_2$ is used to represent the mixture of two distributions $\mathcal{P}_1$ and $\mathcal{P}_2$.

**Problem 1.** *Given $T$ samples $(s_t, a_t, s_{t+1}, y_t(s_t, a_t)), t = 0, \ldots, T - 1$ from the adversarially corrupted reward model described above, and a prescribed failure probability $\delta \in (0, 1)$, our goal is to generate a robust estimate $Q_T$ of the optimal value function $Q^*$, and quantify a bound on the $\ell_\infty$-error $\|Q_T - Q^*\|_\infty$ that holds with probability at least $1 - \delta$.*

Specifically, we ask: (i) Can one still hope to (nearly) preserve the optimal $\tilde{O}(1/\sqrt{T})$ rate of vanilla Q-learning? (ii) What are the fundamental limits on performance imposed by the reward-corrupted attack model? As far as we are aware, despite the popularity of Q-learning, answers to neither of these basic questions are available in the literature. The main contribution of our work is to close this gap by developing an algorithm that achieves near-optimal guarantees for the posed problem.

**Challenges.** There are several unique technical challenges in our problem. First, the heavy-tailed nature of the true reward distribution makes it harder for the learner to distinguish between true samples drawn from the tails of such distributions and adversarial outliers. This uncertainty is further exacerbated when the learner has no knowledge at all about the statistics of the reward distributions - a setting we analyze in Section 4. Second, data in our setting are collected in an online, asynchronous manner, where only a single state-action pair is visited at each time-step. Even in the absence of corruption, such a setting is non-trivial to analyze in the non-asymptotic regime. Third, the data is generated based on a time-correlated Markov chain, making it hard to directly apply standard results from robust statistics that deal with i.i.d. data collected offline. As we will discuss throughout the paper, overcoming these challenges requires significant algorithmic and technical innovations.

Before we introduce our proposed approach, let us state an assumption that is standard in the analysis of RL algorithms [39, 42, 43, 18, 19].

**Assumption 1.** *The Markov chain induced by the behavior policy $\mu$ is aperiodic and irreducible.*

If $\pi$ is the stationary distribution of the Markov chain induced by $\mu$, then the above assumption ensures that $\pi(s) > 0, \forall s \in \mathcal{S}$. At stationarity, note that the visitation probability of a particular state-action pair $(s, a)$ is given by $\lambda(s, a) := \pi(s)\mu(a|s)$, which is non-zero, based on our assumptions on the behavior policy. For later use, we further define the *minimum visitation probability* as $\lambda_{\min} = \min_{(s,a) \in \mathcal{S} \times \mathcal{A}} \lambda(s, a)$. To clearly explain our main ideas, we will assume in Sections 3 and 4 that at each time-step $t$, the state $s_t$ is sampled *independently* from its stationary distribution $\pi$. Later, in Section 4.1, we will relax this i.i.d. assumption, and consider single-trajectory Markov data.

## 3 Robust Asynchronous Q-Learning Algorithm (`Robust Async-Q`)

In this section, we develop a robust variant of the Q-learning algorithm that accounts for asynchronously sampled data, and adversarially corrupted rewards. Our algorithm, titled `Robust Async-Q`, is formally described in Algorithm 1. We start by providing an overview of `Robust Async-Q`, and then flesh out the details. Our approach has two core components: (i) **Robust Reward Estimation.** The first main idea is to use the history of reward observations for each state-action pair $(s, a)$ to generate a robust estimate of the mean reward $R(s, a)$; for this purpose, we exploit the univariate trimmed mean estimator from [12]. (ii) **Adaptive Thresholding.** To account for rare events where robust estimation guarantees may not hold, we carefully design an adaptive thresholding mechanism to discard extreme estimates and ensure that the iterates of `Robust Async-Q` remain uniformly bounded. We will show later that by carefully stitching together these ideas, `Robust Async-Q` is able to achieve near-optimal convergence rates. We now supply the details.

• **Idea 1: Reward Filtering Mechanism.** We start by briefly describing the robust univariate trimmed mean estimator from [12] that we will employ for estimating reward functions. Consider a data set $\mathcal{D}$ comprising of $M$ i.i.d. samples of a scalar random variable $X$ with mean $\mu_X$ and variance $\sigma_X^2$. An adversary arbitrarily perturbs up to $\varepsilon M$ of the samples within $\mathcal{D}$ to produce a corrupted data set $\tilde{\mathcal{D}}$; here, $\varepsilon \in [0, 1/2)$ is the fraction of corrupted data. Using $\tilde{\mathcal{D}}$, the corruption fraction $\varepsilon$, and a confidence parameter $\delta$ as inputs, the trimmed mean estimator from [12] produces a robust estimate $\hat{\mu}_X$ of the mean $\mu_X$ in the following way. The data set $\tilde{\mathcal{D}}$ is divided into two equal parts of $M/2$ samples each. The first part is used to compute empirical quantiles for filtering out extreme values. The estimate $\hat{\mu}_X$ is then simply an average of only those data samples in the second part that fall within the computed quantiles. To apply the estimator from [12] in our context, we need to make minor modifications to the algorithm and the analysis in [12] to account for the Huber contamination model introduced in Section 2. The details of these modifications, along with the manner in which the quantiles are computed, are provided in Appendix B. Let $\hat{\mu}_X = \texttt{TRIM}[\tilde{\mathcal{D}}, \varepsilon, \delta]$ be used to succinctly represent the output of the trimmed mean estimator described above. The following result, adapted from [12], will be of use to us in the sequel.

**Theorem 1.** *Given any $\delta \in (0, 1)$, the following holds with probability at least $1 - \delta$,*

$$|\hat{\mu}_X - \mu_X| \leq \mathcal{C} \sigma_X \left( \sqrt{\varepsilon} + \sqrt{\frac{\log(8/\delta)}{M}} \right), \tag{2}$$

*where $\mathcal{C} \geq 1$ is a universal constant.*

To make use of the estimator explained above, our algorithm maintains a reward history for each state-action pair $(s, a) \in \mathcal{S} \times \mathcal{A}$ via a dynamic array $\mathcal{D}_t(s, a)$ that is initialized from the empty set, i.e., $\mathcal{D}_0(s, a) = \emptyset, \forall(s, a)$. Now, under the asynchronous i.i.d. sampling model, at each time-step $t$, the learner observes a fresh state-action pair sampled as $s_t \sim \pi$ and $a_t \sim \mu(\cdot|s_t)$. If $(s, a) = (s_t, a_t)$, the observed reward $y_t(s_t, a_t)$ is appended to the corresponding array $\mathcal{D}_t(s_t, a_t)$. If $(s, a) \neq (s_t, a_t)$, then the corresponding array remains unchanged from before. Using the dynamic data set $\mathcal{D}_t(s_t, a_t)$, the corruption fraction $\varepsilon$, and a confidence level $\delta_1 = \delta/4T$, a robust estimate $\bar{r}_t(s_t, a_t)$ of the true expected reward $R(s_t, a_t)$ is computed as follows: $\bar{r}_t(s_t, a_t) = \texttt{TRIM}[\mathcal{D}_t(s_t, a_t), \varepsilon, \delta_1]$. Here, note that if we wish the overall output of `Robust Async-Q` to be accurate with a prescribed probability of at least $1 - \delta$, then the failure probability $\delta_1 = \delta/(4T)$ that needs to be fed to the trimmed mean estimator needs to be much finer. The operations above are described in lines 4-6 of Algorithm 1.

• **Idea 2: Adaptive Thresholding.** There are two main obstacles that prevent us from directly using $\bar{r}_t(s_t, a_t)$ (as estimated above) as a proxy for the true mean $R(s_t, a_t)$. First, during the initial phases of our algorithm, one may simply not have visited a particular state-action pair enough times for the robust estimation guarantee to be meaningful. *Thus, we need to wait long enough to acquire adequate observations for every state-action pair.* Second, even when each state-action pair has been visited several times, the guarantees associated with the mean estimator from [12] only hold with *high-probability, not deterministically* (as is evident from Theorem 1). As a result, one cannot rule out extreme events, where the output of the trimmed mean estimator can deviate arbitrarily from the true mean. On such events, using $\bar{r}_t(s_t, a_t)$ directly can lead to uncontrolled errors. The above discussion suggests that *robust estimation is insufficient on its own*. To overcome the two issues described above, we introduce the idea of an **adaptive threshold** that dynamically keeps track of the "typical region" where we expect the output of the trimmed mean estimator to lie within. If the estimate $\bar{r}_t(s_t, a_t)$ falls outside this region, we deem it to be "extreme" and simply discard it by thresholding it to 0.

---

**Algorithm 1** Robust Asynchronous Q-Learning Algorithm (`Robust Async-Q`)

---

1: **Input:** Step-size $\alpha$, corruption fraction $\varepsilon$, confidence level $\delta$, iteration count $T$.
2: Initialize datasets $\mathcal{D}_0(s, a) = \emptyset$, for all $(s, a) \in \mathcal{S} \times \mathcal{A}$, and Q-table $Q_0 = 0$.
3: **for** iteration $t = 0, \ldots, T - 1$ **do**
4:      Observe data tuple $\{s_t, a_t, s_{t+1}\}$, and reward $y_t(s_t, a_t)$.
5:      Append $y_t(s_t, a_t)$ to $\mathcal{D}_t(s_t, a_t)$, and compute $\bar{r}_t(s_t, a_t) \leftarrow \text{TRIM}[\mathcal{D}_t(s_t, a_t), \varepsilon, \delta_1]$.
6:      **if** $|\bar{r}_t(s_t, a_t)| > G_t$ **then**
7:          Set $\tilde{r}_t(s_t, a_t) \leftarrow 0$
8:      **else**
9:          Set $\tilde{r}_t(s_t, a_t) \leftarrow \bar{r}_t(s_t, a_t)$
10:      **end if**
11:      Update $Q_{t+1}$ using Eq. (5).
12: **end for**

---

To formally introduce the adaptive threshold, we first define a burn-in time $\bar{T}$ as follows:

$$\bar{T} = \left\lceil \frac{104}{3\lambda_{\min}} \log\left(\frac{8|\mathcal{S}||\mathcal{A}|T}{\delta_1}\right) \right\rceil, \tag{3}$$

where recall from Section 2 that $\lambda_{\min} > 0$ is the minimum state-action visitation probability. Our analysis will reveal that for $\forall t \geq \bar{T}$, the number of visits to each state-action pair $(s, a)$ up to time $t$ is well concentrated around its mean value $\lambda(s, a)t$ with high probability; this is needed to address the first issue of acquiring enough data. We now define our adaptive threshold $G_t$ as follows:

$$G_t = \begin{cases} 0, & \text{if } t \leq \bar{T}, \\ \mathcal{C}\tilde{\sigma}\left(\sqrt{\frac{4\log(8/\delta_1)}{3\lambda_{\min}t}} + \sqrt{\varepsilon}\right) + \tilde{\sigma}, & \text{if } t > \bar{T}, \end{cases} \tag{4}$$

where $\mathcal{C}$ is the universal constant from Theorem 1, and $\tilde{\sigma} = \max\{\bar{R}, \bar{\sigma}\}$; here, note that we implicitly assume $\tilde{\sigma}$ is known, an assumption we will relax later in Section 4. With the threshold $G_t$ in hand, we account for extreme events as follows: if $|\bar{r}_t(s_t, a_t)| > G_t$, then we discard the estimate by thresholding it to 0. Else, we accept the output of the trimmed mean estimator as is. This operation is described in lines 7-11 of Algorithm 1, where the output of the thresholding scheme is denoted by $\tilde{r}_t(s_t, a_t)$. We emphasize here that the design of the adaptive threshold is the most innovative part of our algorithm **and needs to be done "just right" to achieve near-optimal guarantees**: *if the threshold is too tight, then we will reject estimates unnecessarily; if it is too loose, we might end up accepting extreme estimates.* Either of these scenarios can lead to vacuous bounds.

● **Proposed Robust Q-Update.** We can now formally state the update rule of `Robust Async-Q` which uses $\tilde{r}_t(s_t, a_t)$ - as generated above - as a proxy for the true reward mean $R(s_t, a_t)$ in the Q-learning rule of Watkins [1]:

$$Q_{t+1}(s, a) = \begin{cases} (1 - \alpha)Q_t(s, a) + \alpha\left[\tilde{r}_t(s, a) + \gamma \max_{a' \in \mathcal{A}} Q_t(s_{t+1}, a')\right], & \text{if } (s, a) = (s_t, a_t), \\ Q_t(s, a), & \text{if } (s, a) \neq (s_t, a_t). \end{cases} \tag{5}$$

The update rule above ensures that only robust and bounded reward estimates influence the learning dynamics. In the next section, we will see that the combination of robust filtering and thresholding yields finite-time error bounds for `Robust Async-Q` that gracefully degrade with the corruption level $\varepsilon$, while matching the classical Q-learning rate in the absence of corruption.

### 3.1 Main Results for `Robust Async-Q`

In this section, we provide our first set of results for `Robust Async-Q` with known bounds on reward means and variances. To that end, define $d_t := Q_t - Q^*, \forall t \geq 0$. We then have the following result.

**Theorem 2.** *Suppose Assumption 1 holds, and $T$ satisfies: $T > \max\{\bar{T}, \log(T)/(\lambda_{min}(1 - \gamma))\}$. Given any given $\delta \in (0, 1)$, the output of Algorithm 1 with step-size $\alpha = \frac{\log T}{\lambda_{min}(1 - \gamma)T}$ then satisfies the following bound with probability at least $1 - \delta$:*

$$\|d_T\|_\infty \leq \frac{\|d_0\|_\infty}{T} + \mathcal{O}\left(\frac{\tilde{\sigma}}{(1 - \gamma)^{\frac{5}{2}}} \frac{\log T}{\lambda_{min}^{\frac{3}{2}}\sqrt{T}} \sqrt{\log\left(\frac{|\mathcal{S}||\mathcal{A}|T}{\delta}\right)}\right) + \mathcal{O}\left(\frac{\tilde{\sigma}\sqrt{\varepsilon}}{\lambda_{min}(1 - \gamma)}\right). \tag{6}$$

**Discussion of Theorem 2.** To parse the result from Theorem 2, suppose for the moment that there is no corruption, i.e., $\varepsilon = 0$. The dominant convergence rate from Eq. (6) is then $\tilde{O}(1/((1-\gamma)^{2.5}\sqrt{T}))$, which matches the recent finite-time rates for Q-learning obtained in [44, 18]. Up to polynomial factors in $1/(1-\gamma)$, this rate is known to be minimax optimal [19]. When $\varepsilon \neq 0$, our bound features an additive $O(\sqrt{\varepsilon})$ term that depends only on the small corruption fraction $\varepsilon$, *but crucially is not affected by the magnitude of the injected attacks, highlighting the effectiveness of Algorithm 1 in mitigating adversarial influences.* The corruption-induced term is inflated by the noise variance (as one might expect), and by the inverse of the smallest visitation probability $\lambda_{\min}$. Intuitively, poisoning the data for the least-visited state-action pair can make it harder for the learner to reliably estimate the mean reward for this pair. This intuition is formalized by our upper-bound. The main takeaway from Theorem 2 is that despite corruption, `Robust Async-Q` is able to nearly recover the performance of vanilla Q-learning, up to a small $O(\sqrt{\varepsilon})$ term. To our knowledge, **this is the first result on the adversarial robustness of Q-learning under asynchronous sampling.**

**Fundamental Lower Bound.** One might ask: Is the additive $O(\sqrt{\varepsilon})$ term in (6) unavoidable for our problem of interest? We now show that this is indeed the case by establishing an information-theoretic lower bound. To do so, it suffices to consider a simpler *synchronous* observation model [45–47] for the learner, where it gets to observe data for **every** state-action pair $(s,a) \in \mathcal{S} \times \mathcal{A}$ at each time-step $t$. More precisely, in each iteration $t$, we toss a biased coin with probability of heads $1 - \varepsilon$, independently of the past. If the coin lands heads, for each $(s,a) \in \mathcal{S} \times \mathcal{A}$, the learner observes $y_t(s,a) \sim \mathcal{R}(s,a)$. If it lands tails, for each $(s,a) \in \mathcal{S} \times \mathcal{A}$, $y_t(s,a) \sim \mathcal{Q}$, where recall that $\mathcal{Q}$ is an *arbitrary* adversarial distribution. Let us use $\mathcal{H}(\varepsilon, \bar{\sigma}, \mathcal{Q})$ to collectively represent the set of all MDPs and observation models with finite state and action spaces, where the true underlying reward distributions have bounded mean rewards and variance at most $\bar{\sigma}^2$, and the observed rewards are generated based on the synchronous Huber contamination model described above. With a slight abuse of notation, we will use $Q^* \in \mathcal{H}(\varepsilon, \bar{\sigma}, \mathcal{Q})$ to imply that $Q^*$ is the optimal value function of an MDP consistent with the class of MDPs contained in $\mathcal{H}$. Now, suppose the learner is presented with $T$ independent data sets $\tilde{D}_1, \ldots, \tilde{D}_T$, where $\tilde{D}_t = \{y_t(s,a)\}_{(s,a)\in\mathcal{S}\times\mathcal{A}}$. An estimator $\hat{Q}_T$ of $Q^*$ is some measurable function of these $T$ sets. We then have the following *fundamental* lower bound.

**Theorem 3.** *(**Lower Bound**) There exists a universal constant $\tilde{c} > 0$ such that*

$$\inf_{\hat{Q}_T} \sup_{Q^* \in \mathcal{H}(\varepsilon, \bar{\sigma}, \mathcal{Q})} \mathbb{P}\left( \|\hat{Q}_T - Q^*\|_\infty \geq \frac{\tilde{c}\bar{\sigma}\sqrt{\varepsilon}}{(1-\gamma)} \right) \geq \frac{1}{2}.$$

**Main Takeaway.** From the above result, we infer that the additive corruption term in (6) is tight in its dependence on the corruption fraction $\varepsilon$, the discount factor $\gamma$, and the noise variance $\bar{\sigma}$. Interestingly, these dependencies persist even when the learner is presented with a more favorable observation model where it gets to observe rewards for all the state-action pairs simultaneously at each time-step. We note that similar additive corruption terms have been proven to be unavoidable in prior works on robust mean estimation [48–51], and multi-armed bandits with reward corruptions [5, 7, 8]. Our work is the first to show that such a term is also unavoidable for Q-learning. **Collectively, Theorems 2 and 3 establish the near-optimality of our proposed approach, and paint a fairly complete picture for the theme of adversarial robustness in Q-learning.** To complete this picture, one would need to establish a lower bound that also clarifies the dependence on the minimum visitation probability $\lambda_{\min}$. We conjecture that some dependence on $\lambda_{\min}$ is likely unavoidable; however, verifying this formally is left for future work.

Having established the near-optimality of our approach, the next two sections of the paper are devoted to further generalizing our results to scenarios where bounds on the reward means and variances are unknown (Section 4), and when the data is sampled in a Markovian manner (Section 4.1). Before jumping into these sections, we provide brief proof sketches for Theorems 2 and 3.

**Proof Sketch of Theorem 2.** Using the update rule in (5), we start by writing down a recursion for the error $d_t = Q_t - Q^*$ that features two main terms: a noise term that exhibits a martingale difference structure, and a term that captures the effect of adversarial corruption. *The main challenge in the analysis arises from the fact that these two terms are coupled*; notably, this difficulty does not arise when one analyzes the standard Q-learning algorithm. The coupling is a consequence of the fact that the noise term involves the iterate $Q_t$ which, in turn, is affected by the adversarially corrupted reward observations. Our proof strategy is to first control the effect of adversarial corruption via the following lemma, which is the key new tool in our overall analysis.

**Lemma 1.** *(**Bounding Adversarial Effects**) Suppose Assumption 1 holds. With probability at least $1 - \delta/2$, the following items are true for all $t > \bar{T}$: (i) $\tilde{r}_t(s_t, a_t) = \bar{r}_t(s_t, a_t)$, and*

$$(ii) \quad |\tilde{r}_t(s_t, a_t) - R(s_t, a_t)| \leq \mathcal{O}\left(\tilde{\sigma}\left(\sqrt{\frac{\log(8/\delta_1)}{\lambda_{min}t}} + \sqrt{\varepsilon}\right)\right).$$

Lemma 1 tells us that after the burn-in time $\bar{T}$ is passed, with high-probability, no thresholding will take place, i.e., $\tilde{r}_t(s_t, a_t) = \bar{r}_t(s_t, a_t)$, and the reward proxies that we plug into our update rule (5) will be sufficiently accurate estimates of the true reward functions. The main difficulty in establishing Lemma 1 is that the number of times each state-action pair has been visited up to any time-step $t$ is a *random variable*. As such, we first use Bernstein's inequality to create a "good event" on which, after time $\bar{T}$, each state-action pair is sufficiently visited. We then carefully condition on this event to exploit the bound in (2). Lemma 1 helps us control the effect of adversarial corruption. To control the noise term, we first use the adaptive thresholding idea and an inductive argument to establish that the iterate sequence $\{Q_t\}$ generated by `Robust Aysnc-Q` is uniformly bounded, and then apply Azuma-Hoeffding. The complete details of the proof are deferred to Appendix C.

**Proof Sketch of Theorem 3.** The proof of this result relies on carefully constructing two different MDPs and associated attack distributions, such that (i) the optimal Q-functions in the two MDPs differ in magnitude by $\Omega(\bar{\sigma}\sqrt{\varepsilon}/(1 - \gamma))$; and (ii) the distributions of the observed reward samples in the two MDPs are indistinguishable to a learner. The details are provided in Appendix D.

## 4 Reward-Agnostic Robust Asynchronous Q-Learning (`Robust Async-RAQ`)

In the previous section, we developed a robust variant of the asynchronous Q-learning algorithm (`Robust Async-Q`) that achieves near-optimal guarantees under reward corruption, while assuming access to upper bounds on just the first two moments of the true reward distributions. These assumptions enabled us to precisely design the adaptive threshold $G_t$ in Eq. (4) to safeguard against adversarial outliers. We now ask: *Is it possible to preserve the same rates as before while assuming no prior knowledge at all about the reward statistics?* This is a challenging question motivated by real-world applications where precise bounds on the moments of the reward distributions may not be available to the learner. The lack of knowledge of the parameter $\tilde{\sigma} = \max\{\bar{R}, \bar{\sigma}\}$, which previously played a central role in designing the threshold function $G_t$, now creates more uncertainty for the learner to contend with. Nonetheless, we establish that one continue to enjoy the same bounds as before with two simple modifications to Algorithm 1 that we now describe.

`Modification 1` (**Reward Agnostic Threshold**). Our key idea is to use a polynomial function of time, denoted by $m(t) = t^p$, as a proxy for the *unknown* upper-bound $\tilde{\sigma}$. Any positive integer $p \geq 1$ will suffice for our purpose; we will comment on the choice of $p$ shortly. The new threshold is

$$\tilde{G}_t = 0 \text{ if } t \leq \bar{T}; \quad \tilde{G}_t = \mathcal{C}\, m(t)\left(\sqrt{\frac{4\log(8/\delta_1)}{3\lambda_{\min}t}} + \sqrt{\varepsilon}\right) + m(t) \text{ if } t > \bar{T}, \tag{7}$$

where the universal constant $\mathcal{C}$ and the burn-in time $\bar{T}$ are defined as before in Section 3. The intuition for this proxy is quite simple: since $\tilde{\sigma}$ is a constant, any growing function of time will eventually dominate $\tilde{\sigma}$, after which point, the new threshold $\tilde{G}_t$ will serve as an upper-bound for the threshold $G_t$ that we designed earlier in (4). Lemma 1 will kick in at this point.

`Modification 2` (**Failure Probability Modification**). To make the analysis go through, we will require the failure probability parameter $\delta_1$ that is fed as input to the TRIM function, finer than before: we set $\delta_1 = \delta^2 / \left(512\, |\mathcal{S}|^2 |\mathcal{A}|^2 T^{2p+3}\right)$, where $p$ is the same parameter that appears in $m(t)$. Thus, the overall change to Algorithm 1 involves the new choice of $\delta_1$ in line 5, and the replacement of $G_t$ by $\tilde{G}_t$ in line 6. We call this new reward-agnostic variant `Robust Async-RAQ`.

Our main finite-time result for `Robust Async-RAQ` is as follows.

**Theorem 4.** *Suppose the conditions in Theorem 2 hold. Then, given any $\delta \in (0, 1)$, the output of `Robust Async-RAQ` satisfies the following bound with probability at least $1 - \delta$:*

$$\|d_T\|_\infty \leq \frac{\|d_0\|_\infty}{T} + \mathcal{O}\left(\frac{\tilde{\sigma}^{1+1/2p}}{(1-\gamma)^{\frac{5}{2}}} \frac{\log T}{\lambda_{min}^{\frac{3}{2}}\sqrt{T}}\sqrt{\log\left(\frac{|\mathcal{S}||\mathcal{A}|T}{\delta}\right)}\right) + \mathcal{O}\left(\frac{\tilde{\sigma}\sqrt{\varepsilon}}{\lambda_{min}(1-\gamma)}\right). \tag{8}$$

**Main Takeaway.** Comparing equations (8) and (6), we note that even with no prior knowledge of the reward statistics, `Robust Async-RAQ` is able to remarkably preserve the same near-optimal rates we established before, up to a slight inflation in the dependence on $\tilde{\sigma}$ in the dominant term. This goes on to show the flexibility of our overall framework in accommodating asynchronous sampling, adversarial corruptions, and completely unknown reward statistics. Now, let us comment on the choice of $p$ in the function $m(t)$. Making $p$ larger would lead to a shorter wait time before the modified threshold $\tilde{G}_t$ dominates the true threshold $G_t$, and an improvement in dependence on $\tilde{\sigma}$ in (8). However, a larger $p$ would also imply a smaller failure probability $\delta_1$, which will eventually cause our overall bound to get scaled linearly by $p$, since $\delta_1$ fortunately appears inside a logarithm. Due to the latter fact, up to constant factors, making $p$ large does not degrade our final bound.

**Challenges and Technical Novelty in the Proof of Theorem 4.** In addition to the proof challenges for Theorem 2 we discussed earlier, the modified threshold $\tilde{G}_t$ introduces various new subtleties and technical challenges in the proof, which precludes the use of standard concentration tools used typically in the analysis of RL algorithms. Like before, to exploit the martingale structure of the noise term that shows up in our analysis, we need a uniform bound on $\|Q_t\|_\infty$. While this bound was $\mathcal{O}(1)$ previously, in light of the new threshold, it now becomes on the order of $\mathcal{O}(T^p)$. Using this new upper bound with the standard Azuma-Hoeffding inequality will lead to a vacuously large rate that does not reflect the "typical" behavior of the algorithm. Thus, we need a much more intricate analysis than before. Our key observation is that the iterate sequence $\{Q_t\}$ generated by `Robust Async-RAQ` exhibits an interesting structure: they are bounded by a crude $\mathcal{O}(T^p)$ term deterministically, and a finer $\mathcal{O}(1)$ term with high-probability. This observation does not immediately resolve our problem since we now need a finer version of Azuma-Hoeffding that can exploit the structure identified above. In this regard, some common variants of Azuma-Hoeffding for discrete probability spaces [20] and martingale differences with sub-Gaussian tails [21] are inadequate for our purpose, since the martingale difference in our setting neither belongs to a discrete space nor is sub-Gaussian. Fortunately, we are able to leverage an elegant result from [22] on martingale differences that admit a coarse bound deterministically, and a finer bound with high-probability. This refined variant of Azuma-Hoeffding is the key new tool in our analysis, *and, as far as we are aware, has not appeared before in prior finite-time analysis of RL algorithms.* Thus, the proof of `Robust Async-Q` requires considerable innovation relative to prior work; we defer the details to Appendix E.

## 4.1 Extension to Markovian Sampling

We now explain how our developments can be extended to account for single-trajectory Markovian data. Previously, we assumed that at each time-step $t$, $s_t$ is sampled in an i.i.d. manner from the stationary distribution $\pi$ of the Markov chain induced by the behavior policy $\mu$. We now relax this assumption, and let $s_t$ be the state of this Markov chain at time $t$. It is easy to verify that $Z_t = (s_t, a_t, s_{t+1})$ is also a Markov chain, and that this chain is ergodic based on Assumption 1 [52]. Using this fact, we now propose a simple modification to `Robust Async-RAQ` that ignores certain data points. To explain this modification, let $\Omega$ represent the state space for the Markov chain $\{Z_t\}$, and let $\rho$ be its stationary distribution. Following [23], define $d_{mix}(t) := \sup_{Z \in \Omega} D_{TV}\left(\mathbb{P}(Z_t \in \cdot | Z_0 = Z), \rho\right)$, where $D_{TV}$ is used to represent the total variation distance between probability measures. We now define the mixing time as $\bar{\tau} := \inf\{t | d_{mix}(t) \leq 1/4\}$. Finally, we define a "block" parameter $\tau := \lfloor \ell \bar{\tau} \rfloor$, where $\ell = \lceil \log(2T/\delta)/\log 2 \rceil$. The only modification to `Robust Async-RAQ` is that the agent now uses every $\tau$-th sample, and drops the rest. For this variant (described formally in Appendix F), we have the following result.

**Theorem 5.** *Suppose Assumption 1 holds, and $Z_0 \sim \rho$. Then, given any $\delta \in (0, 1)$, for suitably chosen $\alpha$ and large enough $T$, the following bound holds with probability at least $1 - \delta$:*

$$\|d_T\|_\infty \leq \frac{\|d_0\|_\infty}{T} + \mathcal{O}\left(\frac{\tilde{\sigma}^{1+1/2p}}{(1-\gamma)^{\frac{5}{2}}} \frac{\log T}{\lambda_{min}^{\frac{3}{2}}\sqrt{T}} \sqrt{\tau \log\left(\frac{|\mathcal{S}||\mathcal{A}|T}{\delta}\right)}\right) + \mathcal{O}\left(\frac{\tilde{\sigma}\sqrt{\varepsilon}}{\lambda_{min}(1-\gamma)}\right). \quad (9)$$

**Main Takeaway.** Comparing Theorems 4 and 5, we note that despite Markov sampling, we are able to essentially preserve the same bounds as in the i.i.d. case up to an inflation by a factor of $\sqrt{\tau}$, where $\tau$ captures the mixing time of the Markov chain (up to logarithmic factors). Such an inflation by the mixing time shows up for vanilla Q-learning as well [18]. The assumption that $Z_0 \sim \rho$ is only made to simplify some of the algebra as in prior RL work [43, 23]. Overall, **Theorem 5 establishes the first robustness guarantees for Q-learning with single-trajectory Markovian data.**

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

# A   Additional Literature Survey and Standard Results

In this section, we provide a more detailed discussion of the relevant threads of literature.

1. **Q-Learning.** The Q-learning algorithm was first introduced by Watkins and Dayan in [1]. There is a long line of work that has explored the asymptotic performance of Q-learning algorithms in the limit of infinite samples; see, for instance, [53, 39, 40], using ideas from stochastic approximation theory [53, 54]. A more recent strand of literature has focused on the non-asymptotic analysis of Q-learning and its variants [55, 44, 18, 19], accounting also for function approximation [52]. While we build on some of the techniques in these papers, our work departs from this line of literature by considering the robustness of Q-learning to adversarial perturbations - a topic that has not been explored in the papers mentioned above. For a detailed literature review on Q-learning, we refer the reader to [19].

2. **Stochastic Approximation.** Our work is broadly related to the area of stochastic approximation algorithms in reinforcement learning, which includes Q-learning [1] and TD learning [11] as special cases. As mentioned earlier, the asymptotic theory of such algorithms is rich [42]. Finite-time results, however, are much more recent. Initial finite-time results under the i.i.d. sampling model (that we also consider in this work) were provided in [13, 16, 14, 15, 56]. The extension to the Markov setting was first derived in [43] for a projected TD learning algorithm. The assumption of the projection step was later removed in [57] and [58].

3. **Reward Contamination in Multi-Armed Bandits.** A large body of work has explored the effects of reward contamination on the performance of stochastic bandit problems, both for the unstructured multi-armed bandit (MAB) setting [4, 6, 8, 5, 7], and also for structured linear bandits [26, 27, 25, 28]. The basic premise in these papers is that an adversary can modify the true stochastic reward/feedback on certain rounds; a corruption budget $C$ captures the total corruption injected by the adversary over the horizon $T$. In particular, the authors in [8] study a Huber-contaminated reward model like us, where in each round, with probability $\eta$ (independently of the other rounds), the attacker can bias the reward seen by the learner. A fundamental lower bound of $\Omega(\eta T)$ on the regret is also established in [8]. While our reward contamination model is directly inspired by the above line of work, we emphasize that the stochastic approximation setting we study here fundamentally differs from the bandit problem. As such, our algorithms and proof techniques are also different from the bandit literature.

4. **Robust Statistics.** The study of computing different statistics (e.g., mean, variance, etc.) of a data set in the presence of outliers was pioneered by Huber [2, 3]. Since then, the field of robust statistics has significantly advanced, with more recent work focusing on computationally tractable algorithms in the high-dimensional setting [49, 48, 59, 50, 12, 51]. Our paper builds on this rich line of work and uses it in the context of RL.

## A.1   Useful Facts and Results

In this section, we compile a few useful results that will be used by us throughout the proofs. We start by listing some properties of the Bellman optimality operator $\mathcal{T} : \mathbb{R}^{|\mathcal{S}| \times |\mathcal{A}|} \to \mathbb{R}^{|\mathcal{S}| \times |\mathcal{A}|}$ given by:

$$(\mathcal{T}Q)(s,a) = R(s,a) + \gamma \mathbb{E}_{s' \sim \mathcal{P}(\cdot|s,a)} \left[ \max_{a' \in \mathcal{A}} Q(s',a') \right]. \tag{10}$$

It turns out that the optimal state-action value function $Q^*$ is a fixed point of $\mathcal{T}$, i.e., $\mathcal{T}Q^* = Q^*$. Furthermore, $\mathcal{T}$ is contractive in the $\infty$-norm, a fact that we will exploit in all our main convergence proofs. Formally, the Bellman optimality operator satisfies the following contraction property $\forall Q_1, Q_2 \in \mathbb{R}^{|\mathcal{S}| \times |\mathcal{A}|}$:

$$\|\mathcal{T}Q_1 - \mathcal{T}Q_2\|_\infty \leq \gamma \|Q_1 - Q_2\|_\infty. \tag{11}$$

We also state some useful concentration tools for future use.

**Lemma 2.** (***Bernstein's Inequality***) *If* $X_1, X_2, \ldots, X_N$ *are independent random variables with* $\mathbb{P}(|X_i| \le c) = 1$ *and common mean* $\mu$, *then for any* $\varepsilon > 0$:

$$\mathbb{P}(|\bar{X}_N - \mu| > \varepsilon) \le 2 \exp\left\{ -\frac{N\varepsilon^2}{2\sigma^2 + \frac{2c\varepsilon}{3}} \right\}, \tag{12}$$

*where* $\bar{X}_N = \frac{1}{N} \sum_{i=1}^{N} X_i$ *and* $\sigma^2 = \frac{1}{N} \sum_{i=1}^{N} Var(X_i)$.

**Lemma 3.** (***Azuma-Hoeffding***) *Let* $Z_1, Z_2, Z_3, \ldots$ *be a martingale difference sequence with* $|Z_i| \le c_i$ *for all* $i \in \mathbb{N}$, *where each* $c_i$ *is a positive real. Then, for all* $\lambda \ge 0$:

$$\mathbb{P}\left( \left| \sum_{i=1}^{n} Z_i \right| \ge \lambda \right) \le 2e^{-\frac{\lambda^2}{2\sum_{i=1}^{n} c_i^2}}.$$

## B  Analysis of the Trimmed Mean Estimator under Huber Contamination

---

**Algorithm 2** Univariate Trimmed-Mean Estimator from [12] (`trimSC`)

---

**Require:** Corrupted Dataset $\tilde{\mathcal{D}} = \{X_1, X_2, \ldots, X_M\} = \mathcal{D}_1 \oplus \mathcal{D}_2$, such that $|\mathcal{D}_i|_{i \in \{1,2\}} = M/2$; corruption fraction $\varepsilon$; confidence level $\delta$.

1: Set $\zeta = 8\varepsilon + 24\frac{\log(4/\delta)}{M}$.
2: Let $X_1^* \leq X_2^* \leq \cdots \leq X_{M/2}^*$ represent a non-decreasing arrangement of $\mathcal{D}_1$. Compute quantiles: $\alpha = X_{\zeta M}^*, \quad \beta = X_{(1-\zeta)M}^*$.
3: Define the function $\phi_{\alpha,\beta}(x)$ as

$$\phi_{\alpha,\beta}(x) = \begin{cases} \beta & \text{if } x > \beta \\ x & \text{if } x \in [\alpha, \beta] \\ \alpha & \text{if } x < \alpha \end{cases}$$

4: Compute the `trimmed mean`: $\hat{\mu}_X = (2/M) \sum_{X_i \in \mathcal{D}_2} \phi_{\alpha,\beta}(X_i)$.

---

We start by briefly recalling the strong-contamination data model studied in [12]. Consider a data set $\mathcal{D}$ comprising of $M$ i.i.d. samples of a scalar random variable $X$ with mean $\mu_X$ and variance $\sigma_X^2$. An adversary arbitrarily perturbs up to $\varepsilon M$ of the samples within $\mathcal{D}$ to produce a corrupted data set $\tilde{\mathcal{D}}$; here, $\varepsilon \in [0, 1/2)$ is the fraction of corrupted data. Using $\tilde{\mathcal{D}}$, the corruption fraction $\varepsilon$, and a confidence parameter $\delta$ as inputs, the trimmed mean estimator from [12] produces a robust estimate $\hat{\mu}_X$ of the mean $\mu_X$ in the following way. The data set $\tilde{\mathcal{D}}$ is divided into two equal parts of $M/2$ samples each. The first part is used to compute empirical quantiles for filtering out extreme values. The estimate $\hat{\mu}_X$ is then simply an average of only those data samples in the second part that fall within the computed quantiles. Let $\hat{\mu}_X = \text{trimSC}[\tilde{\mathcal{D}}, \varepsilon, \delta]$ be used to succinctly represent the output of the trimmed mean estimator described above, and outlined in Algorithm 2; here, the subscript 'SC' is used to represent the strong contamination attack model considered in [12]. For this setting, we have the following guarantee from [12].

**Theorem 6.** *[12, Theorem 1] Let $\delta \in (0,1)$ be such that $\delta \geq 4e^{-M/2}$, and suppose $\hat{\mu}_X = \text{trimSC}[\tilde{\mathcal{D}}, \varepsilon, \delta]$. Then, there exists an universal constant c, such that with probability at least $1 - \delta$,*

$$|\hat{\mu}_X - \mu_X| \leq c\sigma_X \left( \sqrt{\varepsilon} + \sqrt{\frac{\log(4/\delta)}{M}} \right). \tag{13}$$

Our goal in this section is to show how the same result can be extended to account for the Huber contamination model of interest to us, where each data sample in $\mathcal{D}$ is arbitrarily corrupted with probability $\varepsilon$. For future reference, we will call the Huber-contaminated data set $\mathcal{D}'$. As we will show, all that needs to happen is that Algorithm 2 needs to be invoked with a slightly larger corruption fraction that will follow from our subsequent analysis.

**Step 1. Bounding the number of corrupted samples.** We begin with a dataset $\mathcal{D}$ consisting of $M$ samples, where each sample is independently corrupted with probability $\varepsilon$, as specified in the corruption model described in Section 2. Our first objective is to bound the total number of corrupted samples in this dataset (with high probability). To this end, we define an event $\mathcal{W}$, where the number of corrupted samples does not exceed $3\varepsilon'M/2$, where $\varepsilon'$ is chosen as follows:

$$\varepsilon' = \varepsilon + \frac{32}{3M} \log\left(\frac{4}{\delta}\right). \tag{14}$$

Our goal is to provide an upper bound on the probability of the complementary event $\mathcal{W}^c$. We start by choosing $Y_i$ as an indicator random variable such that $Y_i = 1$ if the $i^{\text{th}}$ sample is corrupted, and $Y_i = 0$ otherwise. Under the Huber contamination model, we have $\mathbb{E}[Y_i] = \varepsilon$ for all $i \in [M]$.

Furthermore, the average variance satisfies $\sum_{i=1}^{M} \text{Var}(Y_i)/M \leq \varepsilon$. Now observe:

$$
\begin{aligned}
\mathcal{W}^{\mathsf{c}} &:= \Big\{ \sum_{i=1}^{M} Y_i \geq \frac{3\varepsilon' M}{2} \Big\} \\
&= \Big\{ \frac{1}{M} \sum_{i=1}^{M} Y_i - \varepsilon \geq \frac{3\varepsilon'}{2} - \varepsilon \Big\} \\
&\implies \Big\{ \frac{1}{M} \sum_{i=1}^{M} Y_i - \varepsilon \geq \frac{\varepsilon'}{2} \Big\},
\end{aligned}
\tag{15}
$$

where in the last step, we used the fact that $\varepsilon' > \varepsilon$. Applying Bernstein's inequality outlined in Lemma 2 yields the following high-probability bound on the event $\mathcal{W}^{\mathsf{c}}$:

$$
\mathbb{P}\left(\mathcal{W}^{\mathsf{c}}\right) \leq 2e^{-\frac{3\varepsilon' M}{32}} \leq \frac{\delta}{2},
\tag{16}
$$

where the last inequality follows from the definition of the inflated corruption fraction $\varepsilon'$ in (14).

**Step 2. Proof of Theorem 1.** To repurpose Algorithm 2 to account for the Huber contamination model, we simply invoke Algorithm 2 with an inflated corruption fraction and a deflated failure probability. Specifically, let $\hat{\mu}_X = \texttt{TRIM}[\mathcal{D}', \varepsilon, \delta] := \texttt{trimSC}[\mathcal{D}', \bar{\varepsilon}, \delta/2]$, where $\bar{\varepsilon} := \frac{3}{2}\varepsilon'$. In simple words, our modified estimation algorithm for the Huber contaminated setting, denoted by $\texttt{TRIM}$, takes as input the Huber-contaminated data set $\mathcal{D}'$, the contamination probability $\varepsilon$, and failure probability $\delta$. It then invokes Algorithm 2 with the same data set, but with an inflated corruption fraction $\bar{\varepsilon}$, and a deflated failure probability $\delta/2$. To analyze the performance of $\hat{\mu}_X$, let us define an event $\mathcal{V}$ as follows:

$$
\mathcal{V} := \left\{ |\hat{\mu}_X - \mu_X| > c\sigma_X \left( \sqrt{\bar{\varepsilon}} + \sqrt{\frac{\log\left(\frac{8}{\delta}\right)}{M}} \right) \right\},
\tag{17}
$$

where $c$ is the universal constant in Theorem 6. We now decompose the event $\mathcal{V}$ as $\mathcal{V} = \{\mathcal{V} \cap \mathcal{W}\} \cup \{\mathcal{V} \cap \mathcal{W}^{\mathsf{c}}\}$, which immediately implies the following:

$$
\begin{aligned}
\mathbb{P}(\mathcal{V}) = \mathbb{P}(\mathcal{V} \cap \mathcal{W}) + \mathbb{P}(\mathcal{V} \cap \mathcal{W}^{\mathsf{c}}) &\leq \mathbb{P}(\mathcal{V} \cap \mathcal{W}) + \mathbb{P}(\mathcal{W}^{\mathsf{c}}) \\
&\leq \mathbb{P}(\mathcal{V}|\mathcal{W}) \cdot \mathbb{P}(\mathcal{W}) + \mathbb{P}(\mathcal{W}^{\mathsf{c}}) \\
&\leq \underbrace{\mathbb{P}(\mathcal{V}|\mathcal{W})}_{(*)} + \underbrace{\mathbb{P}(\mathcal{W}^{\mathsf{c}})}_{(**)}.
\end{aligned}
\tag{18}
$$

From (16), we already know that $(**) \leq \delta/2$. Furthermore, conditioned on the event $\mathcal{W}$, we know that there are at most $\bar{\varepsilon}M$ corrupted samples in the data set $\mathcal{D}'$. Thus, invoking Theorem 6 immediately yields that $(*) \leq \delta/2$. We conclude that with probability at least $1 - \delta$,

$$
|\hat{\mu}_X - \mu_X| \leq c\sigma_X \left( \sqrt{\bar{\varepsilon}} + \sqrt{\frac{\log\left(\frac{8}{\delta}\right)}{M}} \right) \overset{(\bullet)}{\leq} c\sigma_X \left( \sqrt{\frac{3}{2}\varepsilon'} + \sqrt{\frac{\log\left(\frac{8}{\delta}\right)}{M}} \right)
$$
$$
\overset{(\bullet\bullet)}{\leq} \mathcal{C}\sigma_X \left( \sqrt{\varepsilon} + \sqrt{\frac{\log\left(\frac{8}{\delta}\right)}{M}} \right),
\tag{19}
$$

where $\mathcal{C} > c$ is some suitably large universal constant. In $(\bullet)$, we substituted the value of $\bar{\varepsilon}$, while in $(\bullet\bullet)$, we substituted $\varepsilon'$ from Eq. (14), and applied the elementary inequality $\sqrt{a+b} \leq \sqrt{a} + \sqrt{b}$, that holds for all positive scalars $a, b$. The rest follows from simple algebra. We have thus provided a proof for Theorem 1.

# C Proof of Theorem 2

The proof of Theorem 2 follows a careful sequence of arguments that we proceed to outline next. We begin by decomposing the proposed update rule to isolate the key sources of error arising from both adversarial and non-adversarial components. This is followed by establishing $\ell_\infty$-error bounds for the non-adversarial noise in Lemmas [4,5], and for the adversarial corruption in Lemmas [6, 7]. Finally, we complete the proof of Theorem 2 by assembling these results through a simple yet meticulously crafted inductive argument.

**Error Decomposition Step.** First, using the Bellman optimality operator in Eq. (10), the proposed robust Q-Learning update in Eq. (5) is decomposed as follows:

$$Q_{t+1}(s_t, a_t) = (1 - \alpha)Q_t(s_t, a_t) + \alpha \mathcal{T} Q_t(s_t, a_t) + \alpha \eta_t(s_t, a_t). \tag{20}$$

Here, $\eta_t(s_t, a_t)$ is a perturbation that captures the combined effect of noise and adversarial corruption. Specifically, $\eta_t(s_t, a_t)$ is as follows:

$$\eta_t(s_t, a_t) \triangleq \gamma \max_{a' \in \mathcal{A}} Q_t(s_{t+1}, a') - \gamma \mathbb{E}_{s_{t+1} \sim \mathbb{P}(.|s_t, a_t)} \left[ \max_{a' \in \mathcal{A}} Q_t(s_{t+1}, a') \right] + \tilde{r}_t(s_t, a_t) - R(s_t, a_t). \tag{21}$$

To aid the analysis, we further re-define the following two terms which add up to $\eta_t(s_t, a_t)$ in Eq. (21):

$$\eta_{t,1}(s_t, a_t) = \gamma \max_{a' \in \mathcal{A}} Q_t(s_{t+1}, a') - \gamma \mathbb{E}_{s_{t+1} \sim \mathbb{P}(.|s_t, a_t)} \left[ \max_{a' \in \mathcal{A}} Q_t(s_{t+1}, a') \right],$$
$$\eta_{t,2}(s_t, a_t) = \tilde{r}_t(s_t, a_t) - R(s_t, a_t). \tag{22}$$

**Discussion on the Error Terms.** The term $\eta_t(s_t, a_t)$ defined in Equation (21) captures the deviation between the actual and ideal updates for the sampled state-action pair $(s_t, a_t)$ at the $t^{th}$ time step. Under adversarial reward corruption, this deviation naturally decomposes into two components. The first term $\eta_{t,1}(s_t, a_t)$ captures the gap between the noisy Bellman update and the true Bellman update in (10), excluding the reward term. The second term $\eta_{t,2}(s_t, a_t)$ accounts for the difference between the proposed reward proxy and the expected reward. Note that in the absence of corruption, $\tilde{r}_t(s_t, a_t) = r_t(s_t, a_t)$, such that $\mathbb{E}[r_t(s_t, a_t)] = R(s_t, a_t)$. In this case, the entire term $\eta_t(s_t, a_t)$ reduces to the difference between the noisy Bellman update and the true Bellman update.

**Final Error Decomposition and Matrix Formulation.** For aiding our analysis, we now write Eq. (20) in a compact matrix form, by introducing a time-dependent sparse, diagonal matrix $[D_t]_{|\mathcal{S}|^2 . |\mathcal{A}|^2} \triangleq D_t$, whose only non-zero entry corresponds to the sampled state-action pair $(s, a) = (s_t, a_t)$ at the $t^{th}$ iterations, and equals to 1. This allows us to represent the Q-value update for the current state-action pair using matrix notation:

$$Q_{t+1} = (I - \alpha D_t)Q_t + \alpha D_t(\mathcal{T} Q_t) + \alpha \eta_t(s_t, a_t) \mathbb{1}_t, \tag{23}$$

where $\mathbb{1}_t$ is a $|\mathcal{S}|.|\mathcal{A}|$ dimensional indicator vector, which has the value 1 at the position corresponding to $(s_t, a_t)$ and 0 elsewhere. Since, we are concerned with the asynchronous sampling scheme, $D_t$ is a random matrix. As a result, we introduce a new collective error term to account for this randomness, defined as follows:

$$\zeta_t \triangleq \eta_t(s_t, a_t) \mathbb{1}_t - (D_t - D)(Q_t - \mathcal{T} Q_t), \tag{24}$$

where

$$\mathbb{E}_{s_t \sim \pi, a_t \sim \mu(\cdot|s_t)}[D_t] = D, \quad \text{and} \tag{25}$$

$$D = \begin{bmatrix} \lambda(s_1, a_1) & 0 & 0 & \cdots & 0 \\ 0 & \ddots & 0 & \cdots & 0 \\ 0 & 0 & \lambda(s_i, a_i) = \pi(s_i) \cdot \mu(s_i|a_i) & \cdots & 0 \\ \vdots & \vdots & \vdots & \ddots & \vdots \\ 0 & 0 & 0 & \cdots & \lambda(s_{|\mathcal{S}|}, a_{|\mathcal{A}|}) \end{bmatrix}. \tag{26}$$

The definition of $\zeta_t$ in Eq. (24) accounts for the collective vectorized error, which includes the discrepancy described in Eq. (21) as well as the error arising from the asynchronous sampling nature

of the algorithm, captured by the difference $(D_t - D)$. With the introduction of the collective error term in Eq. (24), Eq. (23) can be rewritten as follows:

$$Q_{t+1} = (I - \alpha D)Q_t + \alpha D(\mathcal{T}Q_t) + \alpha \zeta_t. \tag{27}$$

Now, $Q^*$ is the fixed point of the Bellman optimality operator $\mathcal{T}$, as defined in Equation (10), i.e., $\mathcal{T}Q^* = Q^*$. We can leverage this property to construct the error iterates $(Q_t - Q^*)$ as follows:

$$Q_{t+1} - Q^* = (I - \alpha D)(Q_t - Q^*) + \alpha D(\mathcal{T}Q_t - \mathcal{T}Q^*) + \alpha \zeta_t. \tag{28}$$

Unrolling the above recursion over $t + 1$ iterations, we get:

$$Q_{t+1} - Q^* = (I - \alpha D)^{t+1}(Q_0 - Q^*) + \alpha D \sum_{k=0}^{t}(I - \alpha D)^{t-k}(\mathcal{T}Q_k - \mathcal{T}Q^*) + \Delta_t, \tag{29}$$

where $\Delta_t$ is defined as follows:

$$\Delta_t \triangleq \alpha \sum_{k=0}^{t}(I - \alpha D)^{t-k}\zeta_k. \tag{30}$$

Notably, in the presence of adversaries, $\Delta_t$ is not a standard Martingale Difference Sequence (M.D.S) candidate, since adversarial corruptions introduce a new bias term. To isolate the contributions of stochastic noise and adversarial perturbations, we further decompose $\Delta_t$ into two components, $\Delta_{t,1}$ and $\Delta_{t,2}$, such that:

$$\Delta_{t,1} = \alpha \sum_{k=0}^{t}(I - \alpha D)^{t-k}\zeta_{k,1}, \quad \Delta_{t,2} = \alpha \sum_{k=0}^{t}(I - \alpha D)^{t-k}\zeta_{k,2}, \quad \text{where} \tag{31}$$

the noisy $\zeta_{t,1}$ and adversarial $\zeta_{t,2}$ components which contribute to $\zeta_t$ are defined as follows:

$$\zeta_{t,1} \triangleq \eta_{t,1}(s_t, a_t)\mathbb{1}_t - (D_t - D)(Q_t - \mathcal{T}Q_t), \quad \zeta_{t,2} \triangleq \eta_{t,2}(s_t, a_t)\mathbb{1}_t. \tag{32}$$

Also, the $(s, a) - th$ component of the drift parameters in Eq. (31) is denoted as:

$$\Delta_{t,1}(s,a) \triangleq \alpha \sum_{k=0}^{t}(1 - \alpha\lambda(s,a))^{t-k}\zeta_{k,1}(s,a), \quad \Delta_{t,2}(s,a) \triangleq \alpha \sum_{k=0}^{t}(1 - \alpha\lambda(s,a))^{t-k}\zeta_{k,2}(s,a). \tag{33}$$

`Step 1`: **Bounding the Non-Adversarial Noisy Error** $\Delta_{t,1}$. To begin analyzing the overall error, we first consider the contribution from the cumulative non-adversarial noise term $\Delta_{t,1}$, described in Eq. (31). We first argue that $\{\zeta_{k,1}\}_{k\in[t]}$ admits a standard martingale difference sequence (M.D.S). We show this by proving two key properties: uniform boundedness, established in Lemma 4, and the fact that it has a zero conditional expectation, as shown in first part of Lemma 5. In the latter part of Lemma 5, we use the standard Azuma-Hoeffding inequality from Lemma 3 to bound the cumulative error term $\Delta_{t,1}$ arising from the non-adversarial noise. We now proceed to prove the uniform boundedness property in the next result.

**Lemma 4.** *(Bounding Iterates for* `Robust Async-Q`*) The following bounds hold deterministically for all $t \in [T]$:*

$$|\eta_{t,1}(s_t, a_t)| \leq \frac{6\mathcal{C}\tilde{\sigma}}{1 - \gamma}, \quad \|\zeta_{t,1}\|_\infty \leq \frac{12\mathcal{C}\tilde{\sigma}}{1 - \gamma}, \tag{34}$$

*where $\mathcal{C}$ is the universal constant that appears in (4).*

*Proof.* To establish the claimed bounds, our first step is to argue that the iterates generated by `Robust Async-Q` remain uniformly bounded. We will prove the fact via induction. In particular, we claim that for all $s \in \mathcal{S}$, $a \in \mathcal{A}$, and $t \in [T]$, the following is true:

$$|Q_t(s,a)| \leq \frac{3\mathcal{C}\tilde{\sigma}}{1 - \gamma}, \tag{35}$$

where $\mathcal{C}$ is the universal constant in Eq. (4). The base case of induction at $t = 0$ holds trivially since $Q_0(s,a) = 0$ for all $(s,a)$. Now suppose the bound in (35) holds up to time $t$. To show that it also applies to time $t + 1$, notice that for a state-action pair $(s,a) \neq (s_t, a_t)$, $Q_{t+1}(s,a)$ remains unchanged from time $t$ to time $t + 1$, and thus, the induction claim trivially applies to all state-action

pairs that are not sampled at time $t$. Next, for the sampled state-action pair $(s_t, a_t)$ at time $t$, applying the triangle inequality to the asynchronous $Q$-learning update equation in Eq. (5) yields:

$$
\begin{aligned}
|Q_{t+1}(s_t, a_t)| &\leq (1 - \alpha) |Q_t(s_t, a_t)| + \alpha \left| \tilde{r}_t(s_t, a_t) + \gamma \max_{a' \in \mathcal{A}} Q_t(s_{t+1}, a') \right|, \\
&\leq (1 - \alpha) |Q_t(s_t, a_t)| + \alpha \left( |\tilde{r}_t(s_t, a_t)| + \gamma \max_{a' \in \mathcal{A}} |Q_t(s_{t+1}, a')| \right).
\end{aligned} \tag{36}
$$

To proceed, we note from the thresholding operation in lines [6-9] of Algorithm 1 that: $|\tilde{r}_t(s_t, a_t)| \leqq G_t, \forall t \geq 0$. Moreover, from the definition of $G_t$ in Eq. (4), we observe that $G_t = 0$ for all $t \leq \bar{\bar{T}}$. Also, for all $t > \bar{T}$, we further have that $G_t \leq 2\mathcal{C}\tilde{\sigma} + \tilde{\sigma} \leq 3\mathcal{C}\tilde{\sigma}$, where we used the fact that $\mathcal{C} \geq 1$, and the definition of $\bar{T}$ in Eq. (3). We thus conclude that in light of the thresholding step in Algorithm 1, the following holds deterministically at all time-steps: $|\tilde{r}_t(s_t, a_t)| \leq 3\mathcal{C}\tilde{\sigma}$. Plugging this bound into Eq. (36), and using the induction hypothesis, we obtain the following for the sampled state-action pair $(s_t, a_t)$ at the $t^{th}$ instant:

$$
\begin{aligned}
|Q_{t+1}(s_t, a_t)| &\leq (1 - \alpha) \cdot \frac{3\mathcal{C}\tilde{\sigma}}{1 - \gamma} + \alpha \left( 3\mathcal{C}\tilde{\sigma} + \gamma \cdot \frac{3\mathcal{C}\tilde{\sigma}}{1 - \gamma} \right), \\
&= \left( \frac{1 - \alpha}{1 - \gamma} + \frac{\alpha}{1 - \gamma} \right) 3\mathcal{C}\tilde{\sigma} \leq \frac{3\mathcal{C}\tilde{\sigma}}{1 - \gamma}.
\end{aligned}
$$

We have thus shown that the induction claim in Eq. (35) holds for all state-action pairs $(s, a) \in \mathcal{S} \times \mathcal{A}$, and $\forall t \in [T]$. With a deterministic bound on the iterates, we now proceed to bound the non-adversarial deviation term defined in Eq. (22):

$$
\begin{aligned}
|\eta_{t,1}(s_t, a_t)| &= \left| \gamma \max_{a' \in \mathcal{A}} Q_t(s_{t+1}, a') - \gamma \mathbb{E}_{s_{t+1} \sim \mathbb{P}(\cdot | s_t, a_t)} \left[ \max_{a' \in \mathcal{A}} Q_t(s_{t+1}, a') \right] \right|, \\
&\leq \gamma \left| \max_{a' \in \mathcal{A}} Q_t(s_{t+1}, a') \right| + \gamma \mathbb{E}_{s_{t+1} \sim \mathbb{P}(\cdot | s_t, a_t)} \left| \left[ \max_{a' \in \mathcal{A}} Q_t(s_{t+1}, a') \right] \right|, \\
&\leq \gamma \max_{a' \in \mathcal{A}} |Q_t(s_{t+1}, a')| + \gamma \mathbb{E}_{s_{t+1} \sim \mathbb{P}(\cdot | s_t, a_t)} \left[ \max_{a' \in \mathcal{A}} |Q_t(s_{t+1}, a')| \right], \\
&\leq \gamma \frac{6\mathcal{C}\tilde{\sigma}}{1 - \gamma} \leq \frac{6\mathcal{C}\tilde{\sigma}}{1 - \gamma},
\end{aligned}
$$

where the final inequality uses the bound in Eq. (35). Finally, consider the combined deviation term in Eq. (32). For this term, we have

$$
\begin{aligned}
\|\zeta_{t,1}\|_\infty &\leq |\eta_{t,1}(s_t, a_t)| + \|D_t - D\|_\infty \left( \|Q_t\|_\infty + \|\mathcal{T}Q_t\|_\infty \right) \\
&\overset{(a)}{\leq} \frac{6\mathcal{C}\tilde{\sigma}}{1 - \gamma} + \left( \|Q_t\|_\infty + \|\mathcal{T}Q_t\|_\infty \right) \\
&\overset{(b)}{\leq} \frac{12\mathcal{C}\tilde{\sigma}}{1 - \gamma} \triangleq \bar{\Gamma}.
\end{aligned}
$$

In the above steps, for (a), we used the previously established bound on $|\eta_{t,1}(s_t, a_t)|$, along with the fact that $\|D_t - D\|_\infty \leq 1$. For (b), we used (35) to deduce that $\|Q_t\|_\infty$ and $\|\mathcal{T}Q_t\|_\infty$ are both upper-bounded by $\frac{3\mathcal{C}\tilde{\sigma}}{1 - \gamma}$. In particular, the bound on $\|\mathcal{T}Q_t\|_\infty$ also uses the fact that $|R(s, a)| \leq \bar{R} \leq \tilde{\sigma}$. This completes the proof of Lemma 4, establishing deterministic uniform bounds on the non-adversarial noisy sequences $\{\eta_{k,1}\}_{k \in [t]}$, and $\{\zeta_{k,1}\}_{k \in [t]}$. $\qquad \square$

With the above result in hand, we now proceed to prove Lemma 5, which provides an $\ell_\infty$-norm bound on $\Delta_{t,1}$.

**Lemma 5.** (***Bounding the Noise effect in*** `Robust Async-Q`) *With probability at least* $1 - \frac{\delta}{2}$, *the following bound holds simultaneously* $\forall t \in [T]$:

$$
\left\| \sum_{k=0}^{t} \alpha (I - \alpha D)^{t-k} \zeta_{k,1} \right\|_\infty \leq \frac{12\mathcal{C}\tilde{\sigma}}{1 - \gamma} \sqrt{\frac{\alpha}{2\lambda_{min}} \log \left( \frac{4|\mathcal{S}||\mathcal{A}|T}{\delta} \right)}, \tag{37}
$$

*where* $\zeta_{k,1}$ *is as defined in Eq. (32).*

*Proof.* For a fixed state-action pair $(s, a) \in \mathcal{S} \times \mathcal{A}$, we claim that the process $\{\alpha(1 - \alpha\lambda(s,a))^{t-k}\zeta_{k,1}(s,a)\}_{k \in [t]}$ also admits a martingale difference sequence (M.D.S) with respect to an appropriate filtration. To formally verify this property, we choose a filtration $\mathcal{F}_{k-1}$ denoted by the $\sigma$-algebra generated by the observation history up to time $k-1$, that is, $\mathcal{F}_{k-1} := \sigma(\mathcal{O}_i : 0 \leq i \leq k-1)$, where $\mathcal{O}_i := \{s_i, a_i, s_{i+1}, y_i(s_i, a_i)\}$. Let us also define an augmented $\sigma$-algebra $\mathcal{G}_k := \sigma(\mathcal{F}_{k-1}, (s_k, a_k))$, such that $\mathcal{F}_{k-1} \subseteq \mathcal{G}_k$. In Lemma 4, we have established the uniform boundedness of $\zeta_{k,1}(s,a)$ for all $(s,a) \in \mathcal{S} \times \mathcal{A}$, and for all $k \in [t]$. To conclude that $\zeta_{k,1}(s,a)$ is indeed a M.D.S, it remains to show that $\mathbb{E}[\zeta_{k,1}(s,a)|\mathcal{F}_{k-1}] = 0$.

**Conditional Zero-Expectation Property for M.D.S.** To proceed, we start evaluating $\mathbb{E}[\zeta_{k,1}|\mathcal{F}_{k-1}]$ as follows:

$$\mathbb{E}[\zeta_{k,1}|\mathcal{F}_{k-1}] = \mathbb{E}\Big[\Big(\eta_{k,1}(s_k, a_k)\mathbb{1}_k - (D_k - D)(Q_k - \mathcal{T}Q_k)\Big)\Big|\mathcal{F}_{k-1}\Big]$$

$$\overset{(\bullet)}{=} \mathbb{E}\Big[\eta_{k,1}(s_k, a_k)\mathbb{1}_k\Big|\mathcal{F}_{k-1}\Big] - \mathbb{E}\Big[(D_k - D)(Q_k - \mathcal{T}Q_k)\Big|\mathcal{F}_{k-1}\Big] \tag{38}$$

$$\overset{(\bullet\bullet)}{=} \mathbb{E}\Big[\mathbb{E}[\eta_{k,1}(s_k, a_k)\mathbb{1}_k|\mathcal{G}_k]\Big|\mathcal{F}_{k-1}\Big] = [\mathbf{0}]_{|\mathcal{S}|\times|\mathcal{A}|}.$$

In $(\bullet)$, we invoke the linearity property of conditional expectation: for integrable random variables $A$ and $B$, and a filtration $\mathcal{F}$, the following $\mathbb{E}[A + B|\mathcal{F}] = \mathbb{E}[A|\mathcal{F}] + \mathbb{E}[B|\mathcal{F}]$ holds almost surely. In $(\bullet\bullet)$, we observe that $Q_k$ is $\mathcal{F}_{k-1}$-adapted and that the sampling at time $k$ is independent of the past, under the i.i.d. sampling model. Also, $\mathbb{E}[D_k] = D$, as explained in Equation (25), it follows that $\mathbb{E}\big[(D_k - D)(Q_k - \mathcal{T}Q_k)|\mathcal{F}_{k-1}\big] = 0$. We also apply the *tower property* of conditional expectation, which states that for nested $\sigma$-algebras $\mathcal{B}_1 \subseteq \mathcal{B}_2$, we have $\mathbb{E}[\mathbb{E}[X|\mathcal{B}_2]|\mathcal{B}_1] = \mathbb{E}[X|\mathcal{B}_1]$ almost surely. Using this property, we note $\mathbb{E}[\eta_{k,1}(s_k, a_k)\mathbb{1}_k|\mathcal{G}_k] = 0$. Hence, we conclude that $\mathbb{E}[\zeta_{k,1}|\mathcal{F}_{k-1}] = [\mathbf{0}]_{|\mathcal{S}|\times|\mathcal{A}|}$. Consequently, it follows that $\mathbb{E}[\zeta_{k,1}(s,a)|\mathcal{F}_{k-1}] = 0$ for all $(s,a) \in \mathcal{S} \times \mathcal{A}$. Combined with the uniform boundedness of $\zeta_{k,1}(s,a)$ established in Lemma 4, we conclude that $\{\zeta_{k,1}(s,a)\}_{k \in [t]}$ is indeed a uniformly bounded martingale difference sequence (M.D.S).

**Establishing the Final Bound on $\Delta_{t,1}$.** The boundedness and zero conditional expectation of the noise sequence $\{\zeta_{k,1}\}_{k \in [t]}$, as established in Lemma 4 and Eq. (38), respectively, allow us to invoke the Azuma–Hoeffding inequality described in Lemma 3 to control the deviation of the accumulated noise term. Specifically, we aim to bound $\|\Delta_{t,1}\|_\infty$ described in Eq. (31) with high probability. To achieve this, we analyze each component $\Delta_{t,1}(s,a)$ of the vector $\Delta_{t,1}$ and notice that based on Azuma-Hoeffding, for a fixed $(s,a) \in \mathcal{S} \times \mathcal{A}$ and time-step $t \in [T]$, the following high-probability concentration bound holds with probability at least $1 - \bar{\delta}_1$:

$$|\Delta_{t,1}(s,a)| = \Big|\sum_{k=0}^{t} \alpha(1 - \alpha\lambda(s,a))^{t-k}\zeta_{k,1}(s,a)\Big| \overset{(a)}{\leq} \bar{\Gamma}\sqrt{\frac{\alpha^2}{2}\log\Big(\frac{2}{\bar{\delta}_1}\Big)\sum_{k=0}^{t}(1 - \alpha\lambda(s,a))^{2(t-k)}},$$

$$\overset{(b)}{\leq} \bar{\Gamma}\sqrt{\frac{\alpha^2}{2}\log\Big(\frac{2}{\bar{\delta}_1}\Big)\sum_{r=0}^{\infty}(1 - \alpha\lambda(s,a))^{r}},$$

$$\overset{(c)}{\leq} \bar{\Gamma}\sqrt{\frac{\alpha}{2\lambda_{\min}}\log\Big(\frac{2}{\bar{\delta}_1}\Big)},$$

$$\tag{39}$$

where $\bar{\Gamma}$ is as defined in Lemma 4. We use the standard Azuma-Hoeffding inequality in $(a)$. In $(b)$, we substituted the sum of even powers by a dominating infinite sum of natural powers. In $(c)$, we have used the fact that $\lambda(s,a) \geq \lambda_{\min}$ for all $(s,a) \in \mathcal{S} \times \mathcal{A}$. Now, union bounding over all such good events for all state-action pairs $(s,a) \in \mathcal{S} \times \mathcal{A}$, and time-steps $t \in [T]$, we note that the bound derived above *holds simultaneously* for all state-action pairs and all time-steps with probability at least $1 - \bar{\delta}_1|\mathcal{S}||\mathcal{A}|T$.

Next, in order to simplify, we substitute $\bar{\delta}_1 = \delta/(2|\mathcal{S}||\mathcal{A}|T)$, and $\bar{\Gamma} = 12\mathcal{C}\tilde{\sigma}/(1 - \gamma)$. We then obtain that the following also holds for all $t \in [T]$ with probability at least $1 - \frac{\delta}{2}$:

$$\Big\|\sum_{k=0}^{t} \alpha(I - \alpha D)^{t-k}\zeta_{k,1}\Big\|_\infty = \max_{(s,a)\in\mathcal{S}\times\mathcal{A}}\Big|\sum_{k=0}^{t}\alpha(1 - \alpha\lambda(s,a))^{t-k}\zeta_{k,1}(s,a)\Big|$$

$$\leq \frac{12\mathcal{C}\tilde{\sigma}}{1 - \gamma}\sqrt{\frac{\alpha}{2\lambda_{\min}}\log\Big(\frac{4|\mathcal{S}||\mathcal{A}|T}{\delta}\Big)} \triangleq \bar{\Delta}_{t,1}. \tag{40}$$

This completes the proof. □

Step 2: **Bounding the Adversarial Term** $\Delta_{t,2}$. Before discussing the bound on the adversarial noise term $\Delta_{t,2}$ under the asynchronous sampling model, we first fix some notations that will be used frequently in Lemmas 6 and 7. Denote by $\mathcal{N}_t(s,a)$ a random variable which represents the count of the number of times the state-action pair $(s,a)$ has been visited up to (and including) time $t$. Here, $\mathbb{1}_k(s,a)$ denotes the indicator variable that takes the value 1 if the state-action pair $(s_k, a_k)$ at iteration $k$ is equal to $(s,a)$, and 0 otherwise. Thus, we observe the fact that $\mathcal{N}_t(s,a) = \sum_{k \in [t]} \mathbb{1}_k(s,a)$. Under the i.i.d. sampling model, the probability of visiting a particular $(s,a)$ pair at each time-step is given by $\lambda(s,a) = \pi(s)\mu(a|s)$. As a result, the following is true:

$$\mathbb{E}\left[\mathcal{N}_t(s,a)\right] = \lambda(s,a)t. \tag{41}$$

Building on the above fact, we now construct a "good event" of sufficient measure on which, after a burn-in time, the number of visits to each state-action pair will concentrate around its mean value. To that end, we have the following simple application of Bernstein's inequality.

**Lemma 6.** *(**Constructing Good Event**) There exists an event $\mathcal{K}$ of measure at least $1 - \frac{\delta_1}{4}$, on which, the following holds simultaneously $\forall(s,a) \in \mathcal{S} \times \mathcal{A}$, $\forall t \geq \bar{T}$:*

$$\mathcal{N}_t(s,a) \geq \frac{3}{4}\lambda_{\min} \cdot t,$$

*where $\bar{T} = \left\lceil \frac{104}{3\lambda_{\min}} \log\left(\frac{8|\mathcal{S}||\mathcal{A}|T}{\delta_1}\right) \right\rceil$.*

*Proof.* We start by writing $\mathcal{N}_t(s,a) = \sum_{k \in [t]} \mathbb{1}_k(s,a)$, and observing the following basic facts: $\mathbb{E}[\mathbb{1}_k(s,a)] = \lambda(s,a)$, and $\text{Var}[\mathbb{1}_k(s,a))] \leq \lambda(s,a)$. For a fixed $(s,a) \in \mathcal{S} \times \mathcal{A}$ and fixed $t \in T$, the probability of the following event $\mathcal{K}_1^{\complement}(s,a,t) = \{\mathcal{N}_t(s,a) \leq \frac{3}{4}\lambda(s,a)t\}$ can be bounded using Bernstein's inequality:

$$\begin{aligned}
\mathbb{P}(\mathcal{K}_1^{\complement}(s,a,t)) &= \mathbb{P}\left(\left\{\mathcal{N}_t(s,a) \leq \frac{3}{4}\lambda(s,a)t\right\}\right) \\
&\leq \mathbb{P}\left(\left\{\left|\mathcal{N}_t(s,a) - \mathbb{E}\left[\mathcal{N}_t(s,a)\right]\right| \geq \frac{1}{4}\lambda(s,a)t\right\}\right) \leq 2e^{\left(-\frac{3}{104}\lambda(s,a)t\right)}.
\end{aligned} \tag{42}$$

Let us set $2e^{\left(-\frac{3}{104}\lambda(s,a)t\right)} \leq \hat{\delta}$. Thus, for a fixed state-action pair $(s,a) \in \mathcal{S} \times \mathcal{A}$, and a fixed $t \in T$:

$$\mathbb{P}(\mathcal{K}_1(s,a,t)) \geq 1 - \hat{\delta}, \quad \text{provided} \ \ t \geq \frac{104}{3\lambda(s,a)} \log\left(\frac{2}{\hat{\delta}}\right) \triangleq \bar{T}(s,a).$$

Union-bounding over all state-action pairs $(s,a) \in \mathcal{S} \times \mathcal{A}$ and all time-steps $t \geq \max_{(s,a) \in \mathcal{S} \times \mathcal{A}} \bar{T}(s,a)$, we conclude that there exists an event $\mathcal{K}$ of measure at least $1 - \hat{\delta}|\mathcal{S}||\mathcal{A}|T$, on which the following holds simultaneously for all state-action pairs $(s,a) \in \mathcal{S} \times \mathcal{A}$:

$$\mathcal{N}_t(s,a) \geq \frac{3}{4}\lambda(s,a)t \geq \frac{3}{4}\lambda_{\min}t,$$

provided $t \geq \bar{T}$, with $\bar{T}$ as defined in the statement of the lemma with $\hat{\delta} = \delta_1/4|\mathcal{S}||\mathcal{A}|T$. This concludes the proof. □

**Lemma 7.** *(**Bounding Adversarial Corruption in** `Robust Async-Q`) With probability at least $1 - \frac{\delta}{2}$, the following bound holds simultaneously $\forall t \in [T]$:*

$$\left\| \sum_{k=0}^{t} \alpha(I - \alpha D)^{t-k}\zeta_{k,2} \right\|_{\infty} \leq 10\alpha\mathcal{C}\tilde{\sigma}\left(\sqrt{\frac{T}{\lambda_{\min}} \log\left(\frac{32|\mathcal{S}||\mathcal{A}|T^2}{\delta}\right)}\right) + \frac{\mathcal{C}\tilde{\sigma}}{\lambda_{\min}}\sqrt{\varepsilon}, \tag{43}$$

*where $\zeta_{k,2}$ is defined in Eq. (32).*

*Proof.* We will split our analysis into two separate cases.

**Case I**: When $t \leq \bar{T}$, the term on the left-hand side of Eq. (43) deterministically simplifies to:

$$\left\| \sum_{k=0}^{t} \alpha(I - \alpha D)^{t-k} \zeta_{k,2} \right\|_{\infty} \overset{(*)}{\leq} \alpha \bar{R} \bar{T} \overset{(**)}{\leq} \alpha \tilde{\sigma} \cdot \sqrt{\frac{104T}{3\lambda_{\min}} \log\left(\frac{8|\mathcal{S}||\mathcal{A}|T}{\delta_1}\right)},$$

$$\overset{(***)}{\leq} 6\alpha \mathcal{C} \tilde{\sigma} \cdot \sqrt{\frac{T}{\lambda_{\min}} \log\left(\frac{8|\mathcal{S}||\mathcal{A}|T}{\delta_1}\right)}.$$
(44)

In Eq. (44), we leveraged the threshold function described in Eq. (4) to derive the subsequent bound for the case where $k \leq \bar{T}$. It is evident that $\|I - \alpha D\|_{\infty} \leq 1$ and $\|\zeta_{k,2}\|_{\infty} \leq \bar{R} \leq \tilde{\sigma}$, since $\tilde{r}_t(s,a) = 0$ using Eq. (4) for $t \in [\bar{T}]$. Hence, the bound in $(*)$ is satisfied. In $(**)$, we used $\bar{T} \leq \sqrt{\bar{T}}\sqrt{T}$. Finally, we substitute the value of $\bar{T}$ from Eq. (3) to arrive at the final form.[1]

**Case II**: Next, consider the case when $t > \bar{T}$. We start out by considering the following events $\mathcal{E}_k$, and $\mathcal{E}_{k,1}$ for a fixed $k \in [\bar{T}+1, T]$:

$$\mathcal{E}_k \triangleq \left\{ |\bar{r}_k(s_k, a_k) - R(s_k, a_k)| \leq \mathcal{C}\tilde{\sigma} \left( \sqrt{\frac{4}{3} \frac{\log\left(\frac{4}{\delta_1}\right)}{\lambda_{\min} k}} + \sqrt{\varepsilon} \right) \right\}.$$
(45)

$$\mathcal{E}_{k,1} \triangleq \left\{ |\bar{r}_k(s_k, a_k) - R(s_k, a_k)| \leq \mathcal{C}\tilde{\sigma} \left( \sqrt{\frac{\log\left(\frac{4}{\delta_1}\right)}{\mathcal{N}_k(s_k, a_k)}} + \sqrt{\varepsilon} \right) \right\}.$$
(46)

Next, let us borrow the good event $\mathcal{K}$ from Lemma 6, and decompose the complement of the event $\mathcal{E}_k$ described in Eq. (45) as follows:

$$\{\mathcal{E}_k^{\complement}\} := \{\mathcal{E}_k^{\complement}\} \cap \{\mathcal{K} \cup \mathcal{K}^{\complement}\} = \{\mathcal{E}_k^{\complement} \cap \mathcal{K}\} \cup \{\mathcal{E}_k^{\complement} \cap \mathcal{K}^{\complement}\}.$$
(47)

This immediately implies the following:

$$\mathbb{P}(\mathcal{E}_k^{\complement}) = \mathbb{P}(\mathcal{E}_k^{\complement} \cap \mathcal{K}) + \mathbb{P}(\mathcal{E}_k^{\complement} \cap \mathcal{K}^{\complement}),$$
$$\leq \mathbb{P}(\mathcal{E}_k^{\complement} \cap \mathcal{K}) + \mathbb{P}(\mathcal{K}^{\complement}).$$
(48)

From Lemma 6, on the good event $\mathcal{K}$, we know that for $t \geq \bar{T}$, the following holds: $\mathcal{N}_t(s,a) \geq \frac{3}{4}\lambda_{\min} t$ for all state-action pairs $(s,a) \in \mathcal{S} \times \mathcal{A}$. Next, we establish a bound on $\mathbb{P}(\mathcal{E}_k^{\complement} \cap \mathcal{K})$ in Eq. (48) as follows:

$$\mathbb{P}(\mathcal{E}_k^{\complement} \cap \mathcal{K}) = \sum_{j=\frac{3}{4}\lambda_{\min} k}^{k} \mathbb{P}\left(\mathcal{E}_k^{\complement} \cap \mathcal{K} \cap \{\mathcal{N}_k(s_k, a_k) = j\}\right),$$

$$\leq \sum_{j=\frac{3}{4}\lambda_{\min} k}^{k} \mathbb{P}\left(\mathcal{E}_k^{\complement} \cap \{\mathcal{N}_k(s_k, a_k) = j\}\right),$$

$$\leq \sum_{j=\frac{3}{4}\lambda_{\min} k}^{k} \mathbb{P}\left(\mathcal{E}_k^{\complement}|\{\mathcal{N}_k(s_k, a_k) = j\}\right) \cdot \mathbb{P}\left(\{\mathcal{N}_k(s_k, a_k) = j\}\right),$$
(49)

$$\overset{(\bullet)}{\leq} \sum_{j=\frac{3}{4}\lambda_{\min} k}^{k} \mathbb{P}\left(\mathcal{E}_{k,1}^{\complement}|\{\mathcal{N}_k(s_k, a_k) = j\}\right) \cdot \mathbb{P}\left(\{\mathcal{N}_k(s_k, a_k) = j\}\right),$$

$$\overset{(\bullet\bullet)}{\leq} \delta_1 \cdot \sum_{j=\frac{3}{4}\lambda_{\min} k}^{k} \mathbb{P}\left(\{\mathcal{N}_k(s_k, a_k) = j\}\right),$$

$$\overset{(\bullet\bullet\bullet)}{\leq} \delta_1 \cdot \sum_{j=0}^{k} \mathbb{P}\left(\{\mathcal{N}_k(s_k, a_k) = j\}\right) = \delta_1.$$

---

[1] For simplicity, we assume $\bar{T} = \frac{104}{3\lambda_{\min}} \log\left(\frac{8|\mathcal{S}||\mathcal{A}|T}{\delta_1}\right)$ instead of using the exact value $\bar{T} = \left\lceil \frac{104}{3\lambda_{\min}} \log\left(\frac{8|\mathcal{S}||\mathcal{A}|T}{\delta_1}\right) \right\rceil$. This approximation has no significant impact on our analysis.

In ($\bullet$), for any fixed $k \in [\bar{T} + 1, T]$ and $j \in \left[\frac{3}{4}\lambda_{\min}k, k\right]$, the deviation bound specified by the event $\mathcal{E}_k$ in Eq. (45) is looser than that in $\mathcal{E}_{k,1}$ in Eq. (46) conditioned on $\mathcal{N}_k(s_k, a_k) = j$. Specifically, the following is true:

$$\{\mathcal{E}_k^{\mathsf{C}} | \mathcal{N}_k(s_k, a_k) = j\} \implies \{\mathcal{E}_{k,1}^{\mathsf{C}} | \mathcal{N}_k(s_k, a_k) = j\}. \tag{50}$$

In ($\bullet\bullet$), by conditioning on $\mathcal{N}_k(s_k, a_k)$, we eliminate the randomness associated with asynchronous sampling. Since $j \geq \frac{3}{4}\lambda_{\min}k$, and $k \geq \bar{T} \geq T_{\lim} = \left\lceil \frac{8}{3\lambda_{\min}} \log\left(\frac{4}{\delta_1}\right) \right\rceil$ in Case II, it implies that $j \geq \frac{3}{4}\lambda_{\min}T_{\lim} \geq 2\log\left(\frac{4}{\delta_1}\right)$. Hence, when we fix $\mathcal{N}_k(s_k, a_k) = j \in \left[\frac{3}{4}\lambda_{\min}k, k\right]$, we can leverage the robust mean guarantee in Theorem 1 as follows:

$$\mathbb{P}\left(\mathcal{E}_{k,1}^{\mathsf{C}} | \{\mathcal{N}_k(s_k, a_k) = j\}\right) \leq \delta_1. \tag{51}$$

Lastly, in ($\bullet\bullet\bullet$), we used the fact that $\sum_{j=0}^{k} \mathbb{P}\left(\{\mathcal{N}_k(s_k, a_k) = j\}\right) = 1$. With Eq. (49), we can further simplify our decomposition in Eq. (48) as follows:

$$\begin{aligned} \mathbb{P}(\mathcal{E}_k^{\mathsf{C}}) &= \mathbb{P}(\mathcal{E}^{\mathsf{C}} \cap \mathcal{K}) + \mathbb{P}(\mathcal{K}^{\mathsf{C}}), \\ &\overset{(*)}{\leq} \delta_1 + \frac{\delta_1}{4} \leq 2\delta_1. \end{aligned} \tag{52}$$

In step ($*$), we applied the upper bound on the probability of the good event $\mathcal{K}$ established in Lemma 6. Combining these results, we conclude that the following holds for a fixed $k \in [\bar{T} + 1, T]$:

$$\mathbb{P}(\mathcal{E}_k) \geq 1 - 2\delta_1. \tag{53}$$

Union-bounding over all time-steps $k \in [\bar{T} + 1, T]$, we conclude that there exists an event $\mathcal{J}$ of measure at least $1 - 2\delta_1 T$, on which, the following holds simultaneously for all time steps $k \in [\bar{T} + 1, T]$:

$$|\bar{r}_k(s_k, a_k) - R(s_k, a_k)| \leq \mathcal{C}\tilde{\sigma}\left(\sqrt{\frac{4}{3} \cdot \frac{\log\left(\frac{4}{\delta_1}\right)}{\lambda_{\min}k}} + \sqrt{\varepsilon}\right). \tag{54}$$

Now notice that on the good event $\mathcal{J}$ defined as above, when $k > \bar{T}$, the following is true:

$$|\bar{r}_k(s_k, a_k)| \leq \mathcal{C}\tilde{\sigma}\left(\sqrt{\frac{4}{3} \cdot \frac{\log\left(\frac{4}{\delta_1}\right)}{\lambda_{\min}k}} + \sqrt{\varepsilon}\right) + |R(s_k, a_k)| \leq G_k, \tag{55}$$

where we used $|R(s_k, a_k)| \leq \bar{R} \leq \tilde{\sigma}$, and the definition of the threshold $G_k$ from (4). We conclude that on event $\mathcal{J}$, the thresholding step in line 7 of Algorithm 1 will get bypassed, ensuring that $\tilde{r}_k(s_k, a_k) = \bar{r}_k(s_k, a_k), \forall k > \bar{T}$. Crucially, based on (54), this implies that on the event $\mathcal{J}$, the following deviation bound on the reward proxy applies simultaneously for all time steps $k \in [\bar{T} + 1, T]$:

$$|\tilde{r}_k(s_k, a_k) - R(s_k, a_k)| \leq \mathcal{C}\tilde{\sigma}\left(\sqrt{\frac{4}{3} \cdot \frac{\log\left(\frac{4}{\delta_1}\right)}{\lambda_{\min}k}} + \sqrt{\varepsilon}\right). \tag{56}$$

Now, we substitute $\delta_1 = \delta/4T$, ensuring that the event $\mathcal{J}$ takes place with probability at least $1 - \frac{\delta}{2}$. Before moving forward, we pause to note that the aforementioned arguments have already established Lemma 1 in the main text.

In the remainder of the proof, we will condition on the good event $\mathcal{J}$ on which (56) holds. On this event, it is easy to see that for $k > \bar{T}$,

$$\|\zeta_{k,2}\|_\infty = \left\| \left[ \tilde{r}_k(s_k, a_k) - R(s_k, a_k) \right] \mathbb{1}_k \right\|_\infty$$

$$= |\tilde{r}_k(s_k, a_k) - R(s_k, a_k)| \leq \mathcal{C}\tilde{\sigma} \left( \sqrt{\frac{4}{3} \cdot \frac{\log\left(\frac{4}{\delta_1}\right)}{\lambda_{\min} k}} + \sqrt{\varepsilon} \right). \tag{57}$$

Invoking Eq. (57), the following then holds on event $\mathcal{J}$:

$$\left\| \sum_{k=\bar{T}+1}^{t} \alpha(I - \alpha D)^{t-k} \zeta_{k,2} \right\|_\infty \leq \sum_{k=\bar{T}+1}^{t} \alpha \|(I - \alpha D)\|_\infty^{t-k} \cdot \|\zeta_{k,2}\|_\infty$$

$$\overset{(*)}{\leq} \alpha\mathcal{C}\tilde{\sigma} \sum_{k=\bar{T}+1}^{t} (1 - \alpha\lambda_{\min})^{t-k} \left( \sqrt{\frac{4}{3} \cdot \frac{\log\left(\frac{4}{\delta_1}\right)}{\lambda_{\min} k}} + \sqrt{\varepsilon} \right)$$

$$\leq \alpha\mathcal{C}\tilde{\sigma} \left( \sqrt{\frac{4}{3} \frac{\log\left(\frac{4}{\delta_1}\right)}{\lambda_{\min}}} \right) \sum_{k=\bar{T}+1}^{t} \left( \frac{1}{\sqrt{k}} \right) + \sum_{k=\bar{T}+1}^{t} \alpha(1 - \alpha\lambda_{\min})^{t-k} \mathcal{C}\tilde{\sigma}\sqrt{\varepsilon} \tag{58}$$

$$\overset{(**)}{\leq} \alpha\mathcal{C}\tilde{\sigma} \left( \sqrt{\frac{4}{3} \frac{\log\left(\frac{4}{\delta_1}\right)}{\lambda_{\min}}} \right) \int_{k=\bar{T}+1}^{t} \left( \frac{1}{\sqrt{k}} \right) + \frac{\mathcal{C}\tilde{\sigma}}{\lambda_{\min}}\sqrt{\varepsilon}$$

$$\overset{(***)}{\leq} 2\alpha\mathcal{C}\tilde{\sigma} \left( \sqrt{\frac{4}{3} \frac{T}{\lambda_{\min}} \log\left(\frac{4}{\delta_1}\right)} \right) + \frac{\mathcal{C}\tilde{\sigma}}{\lambda_{\min}}\sqrt{\varepsilon}.$$

Using the bound $\|I - \alpha D\|_\infty \leq (1 - \alpha\lambda_{\min})$ and the deviation bound on $\zeta_{k,2}$ from event $\mathcal{J}$, we obtain step $(*)$. The resulting summation is then separated into two terms—one involving $\frac{1}{\sqrt{k}}$ and another involving a constant $\sqrt{\varepsilon}$. The first term is further upper bounded via an integral approximation $(**)$, while the second term is bounded using the geometric sum of the decaying factor $(1 - \alpha\lambda_{\min})^{t-k}$, which sums to at most $1/(\alpha\lambda_{\min})$. Finally, evaluating the integral and using the upper bound $T$ on the total number of iterations yields the bound in step $(***)$.

Next, to obtain the final bound for Case II, we leverage the bound from Case I to obtain the following (on event $\mathcal{J}$) for all $t > \bar{T}$:

$$\left\| \sum_{k=0}^{t} \alpha(I - \alpha D)^{t-k} \zeta_{k,2} \right\|_\infty \leq \left\| \sum_{k=0}^{\bar{T}} \alpha(I - \alpha D)^{t-k} \zeta_{k,2} \right\|_\infty + \left\| \sum_{k=\bar{T}+1}^{t} \alpha(I - \alpha D)^{t-k} \zeta_{k,2} \right\|_\infty$$

$$\overset{(\dagger)}{\leq} 6\alpha\mathcal{C}\tilde{\sigma} \cdot \sqrt{\frac{T}{\lambda_{\min}} \log\left(\frac{8|\mathcal{S}||\mathcal{A}|T}{\delta_1}\right)} + 2\alpha\mathcal{C}\tilde{\sigma} \left( \sqrt{\frac{4}{3} \frac{T}{\lambda_{\min}} \log\left(\frac{16T}{\delta}\right)} \right) + \frac{\mathcal{C}\tilde{\sigma}}{\lambda_{\min}}\sqrt{\varepsilon}$$

$$\overset{(\dagger\dagger)}{\leq} 10\alpha\mathcal{C}\tilde{\sigma} \left( \sqrt{\frac{T}{\lambda_{\min}} \log\left(\frac{32|\mathcal{S}||\mathcal{A}|T^2}{\delta}\right)} \right) + \frac{\mathcal{C}\tilde{\sigma}}{\lambda_{\min}}\sqrt{\varepsilon} \triangleq \bar{\Delta}_{t,2}.$$

$$\tag{59}$$

In $(\dagger)$, we used the bounds obtained for Case I and Case II. In $(\dagger\dagger)$, we simply used the monotonicity of logarithms and substituted $\delta_1 = \delta/4T$. Lastly, combining our separate analyses for Case I and Case II leads to the claim of the lemma. $\qquad\square$

**Finite-Time Rates for** Robust Async-Q **(Proof of Theorem 2)**: Having established Lemmas 4, 5, 6, and 7, we are now ready to proceed with the proof of the bound stated in Theorem 2. First, to build intuition for the nature of the final bound, let us consider Eq. (27) in the absence of any

contributions from noise or adversaries. In this case, the recursion simplifies to the idealized update rule: $Q_{t+1} = (I - \alpha D)Q_t + \alpha D(\mathcal{T}Q_t)$. Subtracting the fixed point $Q^*$, which satisfies $Q^* = \mathcal{T}Q^*$, we obtain the error recursion $Q_{t+1} - Q^* = (I - \alpha D)(Q_t - Q^*) + \alpha D(\mathcal{T}Q_t - \mathcal{T}Q^*)$. Defining $d_t(s, a) := |Q_t(s, a) - Q^*(s, a)|$, and applying the contractiveness of the Bellman optimality operator under the $\infty$-norm, we can then obtain the following for each state-action pair $(s, a) \in \mathcal{S} \times \mathcal{A}$:

$$d_{t+1}(s, a) \leq (1 - \alpha\lambda(s, a))d_t(s, a) + \alpha\gamma\lambda(s, a)\|d_t\|_\infty,$$
$$\leq (1 - \alpha\lambda_{\min}(1 - \gamma)) \|d_t\|_\infty \tag{60}$$

Since this upper bound holds uniformly over all $(s, a) \in \mathcal{S} \times \mathcal{A}$, we conclude:

$$\|d_{t+1}\|_\infty \leq (1 - \alpha\lambda_{\min}(1 - \gamma)) \|d_t\|_\infty. \tag{61}$$

Unrolling this recursion yields the following for all $t \in [T]$:

$$\|d_t\|_\infty \leq (1 - \alpha\lambda_{\min}(1 - \gamma))^t \|d_0\|_\infty. \tag{62}$$

The goal is to now establish a similar recursion for our setting, while accounting for noise and adversarial corruption. To do so, we note that based on Lemma 5 and Lemma 7, there exists an event - say $\mathcal{Y}$ - of measure at least $1 - \delta$, on which, $\|\Delta_{t,1}\|_\infty + \|\Delta_{t,2}\|_\infty \leq \bar{\Delta}_{t,1} + \bar{\Delta}_{t,2} \triangleq \Delta, \forall t \in [T]$, where $\Delta_{t,1}$ and $\Delta_{t,2}$ are as defined in Eq. (31), $\bar{\Delta}_{t,1}$ is as defined in Eq. (40), and $\bar{\Delta}_{t,2}$ is as defined in Eq. (59). As our induction hypothesis, suppose that on the event $\mathcal{Y}$, the following bound holds for all $t \in [T]$:

$$\|d_t\|_\infty \leq (1 - \alpha\lambda_{\mathtt{min}}(1 - \gamma))^t \|d_0\|_\infty + \frac{\Delta}{1 - \gamma}. \tag{63}$$

For $t = 0$, it is trivially true. Suppose the above bound holds for all time-steps up to time-step $t$. To show that it also applies to time-step $t + 1$, let us revisit Eq. (29) and analyze it component-wise. In order to simplify the notation for algebraic decompositions in the subsequent steps, for two given functions $\{Q_1, Q_2\}$ and their corresponding mappings $\{\mathcal{T}Q_1, \mathcal{T}Q_2\}$ under the influence of the Bellman operator, we denote their component-wise difference as:

$$[Q_1 - Q_2](s, a) \triangleq Q_1(s, a) - Q_2(s, a)$$
$$[\mathcal{T}Q_1 - \mathcal{T}Q_2](s, a) \triangleq \mathcal{T}Q_1(s, a) - \mathcal{T}Q_2(s, a). \tag{64}$$

Similarly, we denote the $(s, a)$-th component of $\Delta_t$ defined in Eq. (31), as $\Delta_t(s, a)$. Now, we proceed component wise, where the $(s, a)$-th component of Eq. (29) gives us the following:

$$[Q_{t+1} - Q^*](s, a) = (1 - \alpha\lambda(s, a))^{t+1}[Q_0 - Q^*](s, a)$$
$$+ \alpha\lambda(s, a) \sum_{k=0}^{t}(1 - \alpha\lambda(s, a))^{t-k}[\mathcal{T}Q_k - \mathcal{T}Q^*](s, a) + \Delta_t(s, a). \tag{65}$$

Taking absolute values on both sides of Eq. (65), and substituting $d_t(s, a) = |[Q_t - Q^*](s, a)|$, we get the following form:

$$d_{t+1}(s, a) \leq (1 - \alpha\lambda(s, a))^{t+1}d_0(s, a) + \alpha\gamma\lambda(s, a) \sum_{k=0}^{t}(1 - \alpha\lambda(s, a))^{t-k}\|d_k\|_\infty + |\Delta_t(s, a)|. \tag{66}$$

Now, substituting $|\Delta_t(s, a)| \leq |\Delta_{t,1}(s, a)| + |\Delta_{t,2}(s, a)| \leq \|\Delta_{t,1}\|_\infty + \|\Delta_{t,2}\|_\infty \leq \bar{\Delta}_{t,1} + \bar{\Delta}_{t,2} = \Delta$ and the claim from Eq. (63) into Eq. (66), we get:

$$d_{t+1}(s, a) \leq \underbrace{(1 - \alpha\lambda(s, a))^{t+1}d_0(s, a) + \alpha\gamma\lambda(s, a) \sum_{k=0}^{t}(1 - \alpha\lambda(s, a))^{t-k}(1 - \alpha\lambda_{\mathtt{min}}(1 - \gamma))^k\|d_0\|_\infty}_{(\bullet)}$$

$$+ \underbrace{\alpha\gamma\lambda(s, a) \sum_{k=0}^{t}(1 - \alpha\lambda(s, a))^{t-k}\frac{\Delta}{1 - \gamma} + \Delta,}_{(\bullet\bullet)}$$

$$\overset{(a)}{\leq} (1 - \alpha\lambda_{\mathtt{min}}(1 - \gamma))^{t+1}\|d_0\|_\infty + +\alpha\gamma\lambda(s, a) \sum_{r=0}^{\infty}(1 - \alpha\lambda(s, a))^r\frac{\Delta}{1 - \gamma} + \Delta,$$

$$\leq (1 - \alpha\lambda_{\mathtt{min}}(1 - \gamma))^{t+1}\|d_0\|_\infty + \frac{\Delta}{1 - \gamma}. \tag{67}$$

In $(a)$, for bounding $(\bullet)$, we used the following argument:

$$
\begin{aligned}
(\bullet) &\leq \left[(1 - \alpha\lambda(s,a))^{t+1} + \alpha\gamma\lambda(s,a)(1 - \alpha\lambda(s,a))^t \sum_{k=0}^{t} \left(\frac{1 - \alpha\lambda_{min}(1-\gamma)}{1 - \alpha\lambda(s,a)}\right)^k\right] \|d_0\|_\infty, \\
&= \left[(1 - \alpha\lambda(s,a))^{t+1} + \alpha\gamma\lambda(s,a)\frac{(1 - \alpha(1-\gamma)\lambda_{\mathtt{min}})^{t+1} - (1 - \alpha\lambda(s,a))^{t+1}}{\alpha\left(\lambda(s,a) - (1-\gamma)\lambda_{\mathtt{min}}\right)}\right] \|d_0\|_\infty, \qquad (68) \\
&\leq \left[(1 - \alpha\lambda(s,a))^{t+1} + \alpha\gamma\lambda(s,a)\frac{(1 - \alpha(1-\gamma)\lambda_{\mathtt{min}})^{t+1} - (1 - \alpha\lambda(s,a))^{t+1}}{\alpha\left(\lambda(s,a) - (1-\gamma)\lambda(s,a)\right)}\right] \|d_0\|_\infty, \\
&\leq (1 - \alpha\lambda_{\min}(1-\gamma))^{t+1} \|d_0\|_\infty.
\end{aligned}
$$

For $(\bullet\bullet)$, we have upper bounded the finite-sum by an infinite-sum as follows:

$$
\begin{aligned}
(\bullet\bullet) &= \alpha\gamma\lambda(s,a) \sum_{k=0}^{t} (1 - \alpha\lambda(s,a))^{t-k}\frac{\Delta}{1-\gamma} + \Delta, \\
&\leq \alpha\gamma\lambda(s,a) \sum_{r=0}^{\infty} (1 - \alpha\lambda(s,a))^{r}\frac{\Delta}{1-\gamma} + \Delta \leq \frac{\Delta}{1-\gamma}. \qquad (69)
\end{aligned}
$$

This settles our claim made in Eq. (63). As a result, we conclude that the following holds on event $\mathcal{Y}$:

$$
\begin{aligned}
\|d_T\|_\infty &\leq (1 - \alpha\lambda_{\mathtt{min}}(1-\gamma))^T\|d_0\|_\infty + \frac{\Delta}{1-\gamma}, \\
&\leq e^{-\alpha\lambda_{\mathtt{min}}(1-\gamma)T}\|d_0\|_\infty + \frac{\Delta}{1-\gamma}. \qquad (70)
\end{aligned}
$$

Substituting $\alpha = \frac{\log T}{\lambda_{\mathtt{min}}T(1-\gamma)}$ in the above display, simplifying, and using the fact that $\mathcal{Y}$ has measure at least $1 - \delta$, we conclude that the following holds with probability $1 - \delta$:

$$
\|d_T\|_\infty \leq \frac{\|d_0\|_\infty}{T} + O\left(\frac{\tilde{\sigma}}{(1-\gamma)^{\frac{5}{2}}\,\lambda_{\mathtt{min}}^{\frac{3}{2}}\sqrt{T}}\sqrt{\log\left(\frac{32|\mathcal{S}||\mathcal{A}|T^2}{\delta}\right)} + \frac{\tilde{\sigma}\sqrt{\varepsilon}}{\lambda_{\mathtt{min}}(1-\gamma)}\right). \qquad (71)
$$

This completes our proof.

# D   Proof of Theorem 3

In this section, we prove the lower bound stated in Theorem 3. The proof is based on constructing two carefully designed observation models under a simple synchronous Huber contamination setting outlined in [45–47], where at each round the learner receives corrupted or clean rewards for all state-action pairs simultaneously. We begin by outlining the core intuition before delving into the technical details. We carefully construct two MDPs that satisfy two crucial properties: (i) the optimal state-action value functions corresponding to the constructed MDPs differ by $\Omega(\sqrt{\varepsilon})$, and (ii) under the Huber contamination model, the observed reward distributions are identical across the two MDPs. This setup ensures that no estimator can reliably distinguish between the two MDPs based on the contaminated observations alone, thereby forcing any estimator to incur an error of at least $\Omega(\sqrt{\varepsilon})$ in the worst case. We now proceed to construct this adversarial instance and formalize the argument.

`Step 1` **(MDP Construction).** To construct the lower bound instance, we consider two MDPs that have a single common state $s$ and a single common action $a$, such that the only source of randomness arises from the observed reward for the state-action pair $(s, a)$. Slightly departing from the notation introduced earlier in the prelude to Theorem 3, we use indices $i = 1$ and $i = 2$ to represent objects associated with MDP 1 and MDP 2, respectively. The true noisy reward distributions $\mathcal{R}_1(s, a)$ and $\mathcal{R}_2(s, a)$ associated with MDPs 1 and 2 are as follows:

$$
\mathcal{R}_1(s, a) = \begin{cases} \frac{\bar{\sigma}}{\sqrt{\varepsilon}} & \text{with prob. } \frac{\varepsilon}{4(1-\varepsilon)}, \\ 0 & \text{with prob. } 1 - \frac{\varepsilon}{4(1-\varepsilon)} \end{cases}, \mathcal{R}_2(s, a) = \begin{cases} -\frac{\bar{\sigma}}{\sqrt{\varepsilon}} & \text{with prob. } \frac{\varepsilon}{4(1-\varepsilon)}, \\ 0 & \text{with prob. } 1 - \frac{\varepsilon}{4(1-\varepsilon)} \end{cases} \tag{72}
$$

where $\bar{\sigma} > 0$ is a fixed constant. Let the expected rewards under distributions $\mathcal{R}_1(s, a)$ and $\mathcal{R}_2(s, a)$ be denoted by $R_1$ and $R_2$, respectively. It is straightforward to check that:

$$
R_1 = \frac{\bar{\sigma}\sqrt{\varepsilon}}{4(1-\varepsilon)}, \qquad R_2 = -\frac{\bar{\sigma}\sqrt{\varepsilon}}{4(1-\varepsilon)}. \tag{73}
$$

Additionally, if $r_1(s, a) \sim \mathcal{R}_1(s, a)$ and $r_2(s, a) \sim \mathcal{R}_2(s, a)$, then the variances of these random variables are as follows:

$$
\texttt{Var}(r_1(s, a)) = \texttt{Var}(r_2(s, a)) \leq \frac{\bar{\sigma}^2}{\varepsilon} \cdot \frac{\varepsilon}{4(1-\varepsilon)} = \frac{\bar{\sigma}^2}{4(1-\varepsilon)} < 0.5\bar{\sigma}^2, \tag{74}
$$

where we have used the assumption that $\varepsilon < 0.5$. Thus, each reward model has a finite variance uniformly bounded above by $\bar{\sigma}^2$. Since there is only one state-action pair, the optimal $Q$-value in each MDP is given by:

$$
Q_i^*(s, a) = \frac{R_i}{1 - \gamma}, \quad i \in \{1, 2\}. \tag{75}
$$

`Step 2` **(Construction of Corrupted Observation Models.)** We now construct adversarial reward contaminations following the Huber contamination model. For each MDP $i \in \{1, 2\}$, the observed reward at the state-action pair $(s, a)$ is drawn with probability $1 - \varepsilon$ from the true underlying distribution $\mathcal{R}_i(s, a)$, and with probability $\varepsilon$ from the adversarial distribution $\mathcal{Q}_i$. The distributions $\mathcal{Q}_i, i \in \{1, 2\}$, represent the corruption distributions, as defined below in Eq. (76) and Eq. (77). Now subject to corruption based on these adversarial distributions, let the resulting reward distributions for MDPs 1 and 2 be denoted by $\tilde{R}_1$ and $\tilde{R}_2$, respectively, where $\tilde{R}_i = (1 - \varepsilon)\mathcal{R}_i(s, a) + \varepsilon \mathcal{Q}_i, i = 1, 2$. These resulting distributions can be easily computed, and are shown in Eq. (76) and Eq. (77).

Adversarial and Resulting Distributions for MDP 1.

$$
\mathcal{Q}_1 = \begin{cases} -\frac{\bar{\sigma}}{\sqrt{\varepsilon}} & \text{with probability } 0.5 \\ 0 & \text{with probability } 0.25 \\ \frac{\bar{\sigma}}{\sqrt{\varepsilon}} & \text{with probability } 0.25. \end{cases}, \quad \tilde{\mathcal{R}}_1 = \begin{cases} -\frac{\bar{\sigma}}{\sqrt{\varepsilon}} & \text{with probability } \frac{\varepsilon}{2} \\ 0 & \text{with probability } 1 - \varepsilon \\ \frac{\bar{\sigma}}{\sqrt{\varepsilon}} & \text{with probability } \frac{\varepsilon}{2}. \end{cases} \tag{76}
$$

Adversarial and Resulting Distributions for MDP 2.

$$
\mathcal{Q}_2 = \begin{cases} -\frac{\bar{\sigma}}{\sqrt{\varepsilon}} & \text{with probability } 0.25 \\ 0 & \text{with probability } 0.25 \\ \frac{\bar{\sigma}}{\sqrt{\varepsilon}} & \text{with probability } 0.5. \end{cases}, \quad \tilde{\mathcal{R}}_2 = \begin{cases} -\frac{\bar{\sigma}}{\sqrt{\varepsilon}} & \text{with probability } \frac{\varepsilon}{2} \\ 0 & \text{with probability } 1 - \varepsilon \\ \frac{\bar{\sigma}}{\sqrt{\varepsilon}} & \text{with probability } \frac{\varepsilon}{2}. \end{cases} \tag{77}
$$

Crucially, note that based on our construction above, $\tilde{\mathcal{R}}_1 = \tilde{\mathcal{R}}_2$. As a result, *a learner cannot distinguish between the corrupted reward distributions of the two MDPs.* However, as established in Step 1, the true (uncorrupted) expected rewards under these MDPs differ. Thus, the corresponding true optimal $Q^*$-values also differ, with the following bound:

$$|Q_1^* - Q_2^*| = \frac{|R_1 - R_2|}{1 - \gamma} = \frac{\bar{\sigma}\sqrt{\varepsilon}}{2(1 - \varepsilon)(1 - \gamma)} \geq \frac{\bar{\sigma}\sqrt{\varepsilon}}{2(1 - \gamma)}, \tag{78}$$

where we will henceforth use the simpler notation $Q_i^*(s, a) \triangleq Q_i^*$ for $i \in \{1, 2\}$ in light of the fact that there is only one state-action pair. We now proceed to establish that *any* estimator of the optimal state-action value function must suffer an error of $\Omega\left(\frac{\bar{\sigma}\sqrt{\varepsilon}}{(1 - \gamma)}\right)$ on at least one of the two MDPs.

Step 3 (**Lower Bound on Estimation Error.**) For $i = 1, \ldots, T$, let $(X_i, Y_i)$ be independent pairs of random observations satisfying:

$$\mathbb{P}(X_i = Y_i = -\bar{\sigma}/\sqrt{\varepsilon}) = \frac{\varepsilon}{2},$$

$$\mathbb{P}(X_i = Y_i = 0) = 1 - \varepsilon, \ \mathbb{P}(X_i = Y_i = \bar{\sigma}/\sqrt{\varepsilon}) = \frac{\varepsilon}{2}.$$

Let us note that $X_i$ is distributed as per $\tilde{R}_1$, and $Y_i$ as per $\tilde{R}_2$. Clearly, the following is true: $\mathbb{P}\left(\{X_i\}_{i \in [T]} = \{Y_i\}_{i \in [T]}\right) = 1$. Now, suppose $\hat{R}_T$ and $\hat{Q}_T$ are estimators for the mean rewards and optimal state-action value functions, respectively, in the two MDPs. As we shall see, establishing a fundamental limit on the performance of $\hat{R}_T$ is sufficient to establish a limit on the performance of $\hat{Q}_T$. To see this, start by noting that

$$\max\left\{\mathbb{P}\left(|\hat{R}_T(\{X_i\}_{i \in [T]}) - R_1| > \frac{\bar{\sigma}\sqrt{\varepsilon}}{8(1 - \varepsilon)}\right), \mathbb{P}\left(|\hat{R}_T(\{Y_i\}_{i \in [T]}) - R_2| > \frac{\bar{\sigma}\sqrt{\varepsilon}}{8(1 - \varepsilon)}\right)\right\}$$

$$\overset{(\bullet)}{\geq} \frac{1}{2}\mathbb{P}\left(\left\{|\hat{R}_T(\{X_i\}_{i \in [T]}) - R_1| > \frac{\bar{\sigma}\sqrt{\varepsilon}}{8(1 - \varepsilon)}\right\} \bigcup \left\{|\hat{R}_T(\{Y_i\}_{i \in [T]}) - R_2| > \frac{\bar{\sigma}\sqrt{\varepsilon}}{8(1 - \varepsilon)}\right\}\right)$$

$$\overset{(\bullet\bullet)}{\geq} \frac{1}{2}\mathbb{P}\left(\hat{R}_T(\{X_i\}_{i \in [T]}) = \hat{R}_T(\{Y_i\}_{i \in [T]})\right)$$

$$\overset{(\bullet\bullet\bullet)}{\geq} \frac{1}{2}\mathbb{P}\left(\{X_i\}_{i \in [T]} = \{Y_i\}_{i \in [T]}\right) = \frac{1}{2}. \tag{79}$$

In step ($\bullet$), we use the inequality $\max\{\mathbb{P}(A), \mathbb{P}(B)\} \geq \frac{1}{2}\mathbb{P}(A \cup B)$ that holds for all events $A$ and $B$. Step ($\bullet\bullet$) follows by substituting the expressions for $\{R_i\}_{i \in \{1,2\}}$ as defined in Eq. (73), which ensures that any estimator outputting the same value on both datasets must incur a certain error on at least one. Finally, for step ($\bullet \bullet \bullet$), we used $\mathbb{P}\left(\{X_i\}_{i \in [T]} = \{Y_i\}_{i \in [T]}\right) = 1$.

Using $1/(1 - \varepsilon) > 1$, we then conclude that:

$$\max\left\{\mathbb{P}\left(\left|\hat{R}_T(\{X_i\}_{i \in [T]}) - R_1\right| > \frac{\bar{\sigma}\sqrt{\varepsilon}}{8}\right), \ \mathbb{P}\left(\left|\hat{R}_T(\{Y_i\}_{i \in [T]}) - R_2\right| > \frac{\bar{\sigma}\sqrt{\varepsilon}}{8}\right)\right\} \geq \frac{1}{2}. \tag{80}$$

In light of Eq. (80), we claim that

$$\max\left\{\mathbb{P}\left(|\hat{Q}_T(\{X_i\}_{i \in [T]}) - Q_1^*| > \frac{\bar{\sigma}\sqrt{\varepsilon}}{8(1 - \gamma)}\right), \ \mathbb{P}\left(|\hat{Q}_T(\{Y_i\}_{i \in [T]}) - Q_2^*| > \frac{\bar{\sigma}\sqrt{\varepsilon}}{8(1 - \gamma)}\right)\right\} \geq \frac{1}{2}. \tag{81}$$

The claim essentially follows from the simple observation that if an optimal state-action value-function estimator $\hat{Q}_T$ can accurately estimate both $Q_1^*$ and $Q_2^*$, then one can use such an estimator to construct accurate estimates of both $R_1$ and $R_2$, thereby violating Eq. (80). Formally, to see that Eq. (80) implies Eq. (81), suppose there exists an estimator $\hat{Q}_T$ such that

$$\max\left\{\mathbb{P}\left(|\hat{Q}_T(\{X_i\}_{i \in [T]}) - Q_1^*| > \frac{\bar{\sigma}\sqrt{\varepsilon}}{8(1 - \gamma)}\right), \ \mathbb{P}\left(|\hat{Q}_T(\{Y_i\}_{i \in [T]}) - Q_2^*| > \frac{\bar{\sigma}\sqrt{\varepsilon}}{8(1 - \gamma)}\right)\right\} < \frac{1}{2}. \tag{82}$$

Using $\hat{Q}_T$, construct a reward estimator $\hat{R}_T = (1 - \gamma)\hat{Q}_T$. From Eq. (75), we then immediately have:

$$\max \left\{ \mathbb{P}\left( \left| \hat{R}_T(\{X_i\}_{i \in [T]}) - R_1 \right| > \frac{\bar{\sigma}\sqrt{\varepsilon}}{8} \right), \ \mathbb{P}\left( \left| \hat{R}_T(\{Y_i\}_{i \in [T]}) - R_2 \right| > \frac{\bar{\sigma}\sqrt{\varepsilon}}{8} \right) \right\} < \frac{1}{2}.$$

(83)

This completes the claim and the proof.

# E Proof of Theorem 4

The finite-time performance of `Robust Async-RAQ` is established in Theorem 4. The first step in the proof of this result is an error-decomposition that mirrors Eq. (29) in Section C. The structure of the rest of the proof is similar to that of Theorem 2 in Appendix C. However, there will be some departures that arise from the use of a reward-agnostic threshold function in Eq. (7). We will highlight these points of departure in our subsequent analysis.

**Step 1: Bound on the Adversarial Term** $\Delta_{t,2}$. We begin by analyzing the contribution of the adversarial corruption term, before turning to the non-adversarial noisy component. The latter necessitates a more refined and intricate analysis, as will become evident in the sequel.

**Lemma 8.** *(Bounding Adversarial Corruption in* `Robust Async-RAQ`*) Suppose $\delta_1 \leq \delta/4T$. Then, with probability at least $1 - \delta/2$, the following bound holds simultaneously for all $t \in [T]$:*

$$\left\| \sum_{k=0}^{t} \alpha(I - \alpha D)^{t-k} \zeta_{k,2} \right\|_\infty \leq \mathcal{O}(\alpha\tilde{\sigma}) \left( \tilde{\sigma}^{1/2p}\sqrt{T} + \sqrt{\frac{T}{\lambda_{\min}} \log\left(\frac{|\mathcal{S}||\mathcal{A}|T}{\delta_1}\right)} \right) + \mathcal{O}\left(\frac{\tilde{\sigma}\sqrt{\varepsilon}}{\lambda_{\min}}\right),$$

*where $\zeta_{k,2}$ is defined in Eq. (32).*

*Proof.* Like in our proof of Lemma 7, we divide the analysis into two cases based on the value of $t$. Since the threshold function defined in Eq. (7) is agnostic to the underlying reward statistics, we introduce an auxiliary time-step $\tilde{T} := \max\left\{\tilde{\sigma}^{1/p}, \bar{T}\right\}$, where $\bar{T}$ was previously defined in Eq. (3), and recall that $p$ is the parameter in the function $m(t) = t^p$ that appears in the modified threshold (7).

**Case I:** Consider first the case where $t \leq \tilde{T}$. We further split up this case into two sub-cases: one where $\tilde{T} = \bar{T}$, and the other where $\tilde{T} = \tilde{\sigma}^{1/p}$. We separately analyze these sub-cases below.

- Suppose $\tilde{T} = \bar{T}$, which implies $t \leq \bar{T}$. Then, by the definition of the threshold function in Eq. (7), we have $\tilde{r}_t(s_t, a_t) = 0$. Consequently, just like in Case 1 of Lemma 7, in this case we have $\|\zeta_{t,2}\|_\infty \leq \tilde{\sigma}$.

- Next, when $\tilde{T} = \tilde{\sigma}^{1/p}$, and $t \in [\bar{T}, \tilde{T}]$, we can use the reward-agnostic threshold function defined in Eq. (7) to bound $\|\zeta_{t,2}\|_\infty$. To see how, start by noting that the following is always true deterministically: $|\tilde{r}_t(s_t, a_t)| \leq \tilde{G}_t, \forall t \geq 0$. Using $m(t) = t^p$ in Eq. (7), and the fact that $t \geq \bar{T}$, we note that for $t \in [\bar{T}, \tilde{T}]$, the following is true: $\tilde{G}_t \leq 3\mathcal{C}t^p \leq 3\mathcal{C}\tilde{T}^p = 3\mathcal{C}\tilde{\sigma}$, where in the last step, we used that in this case $\tilde{T} = \tilde{\sigma}^{1/p}$. Thus, for $t \in [\bar{T}, \tilde{T}]$, we have $|\tilde{r}_t(s_t, a_t)| \leq 3\mathcal{C}\tilde{\sigma}$. As a result, we have $\|\zeta_{t,2}\|_\infty = |\tilde{r}_t(s_t, a_t) - R(s_t, a_t)| \leq 3\mathcal{C}\tilde{\sigma} + \bar{R} \leq 4\mathcal{C}\tilde{\sigma}$, since $\mathcal{C} \geq 1$, and $\bar{R} \leq \tilde{\sigma}$.

From our analysis of the two sub-cases above, we conclude that for $t \leq \tilde{T}$, $\|\zeta_{t,2}\|_\infty \leq 4\mathcal{C}\tilde{\sigma}$. Next, we bound the adversarial corruption term $\Delta_{t,2}$ in the $\infty$-norm for all $t \in [\tilde{T}]$ as follows:

$$
\begin{aligned}
\|\Delta_{t,2}\|_\infty &\leq \alpha \left\| \sum_{k=0}^{t} (I - \alpha D)^{t-k} \zeta_{k,2} \right\|_\infty \\
&\overset{(*)}{\leq} \alpha \sum_{k=0}^{\tilde{T}-1} \|(I - \alpha D)^{t-k}\|_\infty \cdot \|\zeta_{k,2}\|_\infty \\
&\overset{(**)}{\leq} 4\mathcal{C}\alpha\tilde{\sigma}\tilde{T}.
\end{aligned}
\tag{84}
$$

In $(*)$, we apply the triangle inequality, followed by the sub-multiplicative property of the $\infty$-norm. In $(**)$, we use the fact that $\|(I - \alpha D)^{t-k}\|_\infty \leq 1$, and that $\|\zeta_{k,2}\|_\infty \leq 4\mathcal{C}\tilde{\sigma}$, as established earlier for Case I. This completes the analysis for Case I.

**Case II:** We now consider the case when $t > \tilde{T}$. Since $\tilde{T} := \max\left\{\tilde{\sigma}^{1/p}, \bar{T}\right\}$, it follows that $t > \tilde{T} \Rightarrow t > \bar{T}$. Now recall from the analysis of Lemma 7 that there exists an event $\mathcal{J}$ of measure at least $1 - 2\delta_1 T \geq 1 - \delta/2$, on which, the following holds simultaneously for all time steps

$t \in [\bar{T}+1, T]$:

$$|\bar{r}_t(s_t, a_t) - R(s_t, a_t)| \leq \mathcal{C}\tilde{\sigma} \left( \sqrt{\frac{4}{3} \cdot \frac{\log\left(\frac{4}{\delta_1}\right)}{\lambda_{\min} t}} + \sqrt{\varepsilon} \right). \tag{85}$$

On this event, we further have that for $t > \bar{T}$: $|\bar{r}_t(s_t, a_t)| \leq G_t$, where $G_t$ is the original threshold defined in (4). While this condition was enough to prevent any thresholding on event $\mathcal{J}$ for $t > \bar{T}$ for `Robust Async-Q`, it does not immediately imply that thresholding will not take place for `Robust Async-RAQ`. The reason for this stems from the fact that in the new algorithm, the modified threshold $\tilde{G}_t$ in (7) can be an under-approximation of $G_t$ during the period $[\bar{T}, \tilde{T}]$. However, for $t > \tilde{T}$, we have $m(t) = t^p > \tilde{T}^p \geq \tilde{\sigma}$, since $\tilde{T} = \max\{\tilde{\sigma}^{1/p}, \bar{T}\}$. As a result, for $t > \tilde{T}$, we have $G_t \leq \tilde{G}_t$. Consequently, on the event $\mathcal{J}$, we have that for all $t > \tilde{T}$, $|\bar{r}_t(s_t, a_t)| \leq G_t < \tilde{G}_t$. Thus, the thresholding operation in line 7 will get bypassed, ensuring that $\tilde{r}_t(s_t, a_t) = \bar{r}_t(s_t, a_t)$, and, as a result, we conclude based on (85) that on event $\mathcal{J}$, for all $t > \tilde{T}$, the following is true:

$$|\tilde{r}_t(s_t, a_t) - R(s_t, a_t)| \leq \mathcal{C}\tilde{\sigma} \left( \sqrt{\frac{4}{3} \cdot \frac{\log\left(\frac{4}{\delta_1}\right)}{\lambda_{\min} t}} + \sqrt{\varepsilon} \right). \tag{86}$$

Based on the above bound, we can proceed to control the adversarial term $\Delta_{t,2}$ as follows:

$$\left\| \sum_{k=0}^{t} \alpha(I - \alpha D)^{t-k} \zeta_{k,2} \right\|_\infty \leq \left\| \sum_{k=0}^{\tilde{T}} \alpha(I - \alpha D)^{t-k} \zeta_{k,2} \right\|_\infty + \left\| \sum_{k=\tilde{T}+1}^{t} \alpha(I - \alpha D)^{t-k} \zeta_{k,2} \right\|_\infty ,$$

$$\leq 4\mathcal{C}\alpha\tilde{\sigma}\tilde{T} + \mathcal{O}(\mathcal{C}\alpha\tilde{\sigma})\sqrt{T\frac{\log(4/\delta_1)}{\lambda_{\min}}} + \mathcal{O}\left(\frac{\mathcal{C}\tilde{\sigma}\sqrt{\varepsilon}}{\lambda_{\min}}\right),$$

$$\leq 4\mathcal{C}\alpha\tilde{\sigma}\sqrt{\tilde{T}} \cdot \sqrt{T} + \mathcal{O}(\mathcal{C}\alpha\tilde{\sigma})\sqrt{T\frac{\log(4/\delta_1)}{\lambda_{\min}}} + \mathcal{O}\left(\frac{\mathcal{C}\tilde{\sigma}\sqrt{\varepsilon}}{\lambda_{\min}}\right),$$

$$\leq 4\mathcal{C}\alpha\tilde{\sigma}\sqrt{\bar{T} + \sigma^{\frac{1}{p}}} \cdot \sqrt{T} + \mathcal{O}(\mathcal{C}\alpha\tilde{\sigma})\sqrt{T\frac{\log(4/\delta_1)}{\lambda_{\min}}} + \mathcal{O}\left(\frac{\mathcal{C}\tilde{\sigma}\sqrt{\varepsilon}}{\lambda_{\min}}\right),$$

$$\leq \mathcal{O}(\mathcal{C}\alpha\tilde{\sigma})\left(\tilde{\sigma}^{1/2p}\sqrt{T} + \sqrt{\frac{T}{\lambda_{\min}}\log\left(\frac{|\mathcal{S}||\mathcal{A}|T}{\delta_1}\right)}\right) + \mathcal{O}\left(\frac{\mathcal{C}\tilde{\sigma}\sqrt{\varepsilon}}{\lambda_{\min}}\right) \triangleq \tilde{\Delta}_{t,2}. \tag{87}$$

For the first step, we stitched together the bounds for Cases I and II, and followed a similar reasoning as in the proof of Lemma 7. Under the assumption $T \geq \tilde{T}$, we further used $\tilde{T} \leq \sqrt{\tilde{T}} \cdot \sqrt{T}$. Finally, we leveraged the definition $\tilde{T} = \max\left\{\bar{T}, \tilde{\sigma}^{1/p}\right\}$, which implies $\tilde{T} \leq \bar{T} + \tilde{\sigma}^{1/p}$, and plugged in the expression for $\bar{T}$ from (3), followed by simplifications. Combining the bounds obtained in **Case I** and **Case II**, we conclude the proof of Lemma 8. $\qquad\square$

`Step 2:` **Bound the Non-Adversarial Noise Term** $\Delta_{t,1}$. We now proceed to the more delicate part of the analysis that involves controlling the effect of noise. Like before, to control the noise effect using a martingale-based argument, we will derive uniform bounds on the iterates generated by `Robust Async-RAQ`. However, as a departure from the analysis in Appendix C, we will derive two sets of bounds: crude bounds that hold deterministically, and finer bounds that hold with high probability. The rationale for this will become clearer soon. We start with the cruder bounds.

**Lemma 9.** *(Coarse Deterministic Bounds on Iterates for* `Robust Async-RAQ`*) The following bounds hold deterministically for all $t \in [T]$:*

$$|\eta_{t,1}(s_t, a_t)| \leq \frac{6\mathcal{C}T^p}{1-\gamma}, \quad \|\zeta_{t,1}\|_\infty \leq \frac{12\mathcal{C}T^p}{1-\gamma}, \tag{88}$$

*where $\mathcal{C}$ is the universal constant that appears in (4).*

*Proof.* The proof is nearly identical to that of Lemma 4, with the only difference arising from the modified threshold function. Let us start by noting that the following is always true deterministically: $|\tilde{r}_t(s_t, a_t)| \leq \tilde{G}_t, \forall t \geq 0$. Now based on the definition of the modified threshold $\tilde{G}_t$ in (7) and $\bar{T}$ in (3), we have that $\tilde{G}_t = 0, \forall t \leq \bar{T}$, and $\tilde{G}_t \leq 3\mathcal{C}t^p \leq 3\mathcal{C}T^p, \forall t > \bar{T}$. As a result, in Robust Async-RAQ, the reward proxy $\tilde{r}_t(s_t, a_t)$ is deterministically bounded at each time step as $|\tilde{r}_t(s_t, a_t)| \leq \tilde{G}_t \leq 3\mathcal{C}T^p, \forall t \in [T]$. Using this fact, and the exact same inductive reasoning as in the proof of Lemma 4, we can show that:

$$\|Q_t\|_\infty \leq \frac{3\mathcal{C}T^p}{1-\gamma}, \forall t \geq 0. \tag{89}$$

Following the same arguments as in Lemma 4, one can then also show that

$$|\eta_{t,1}(s_t, a_t)| \leq \frac{6\mathcal{C}T^p}{1-\gamma}, \forall t \geq 0. \tag{90}$$

Now fix any state-action pair $(s, a)$, and observe that

$$
\begin{aligned}
|\mathcal{T}Q_t(s, a)| &= |R(s, a) + \gamma \mathbb{E}_{s' \sim \mathcal{P}(\cdot|s,a)}[\max_{a' \in \mathcal{A}} Q_t(s', a')]| \\
&\leq |R(s, a)| + \gamma \mathbb{E}_{s' \sim \mathcal{P}(\cdot|s,a)}[|\max_{a' \in \mathcal{A}} Q_t(s', a')|] \\
&\overset{(a)}{\leq} \tilde{\sigma} + \frac{3\gamma\mathcal{C}T^p}{1-\gamma} \\
&\overset{(b)}{\leq} 3\mathcal{C}T^p + \frac{3\gamma\mathcal{C}T^p}{1-\gamma} \\
&= \frac{3\mathcal{C}T^p}{1-\gamma}.
\end{aligned}
\tag{91}
$$

For (a), we used $|R(s, a)| \leq \tilde{\sigma}$ and Eq. (89). For (b), we used the fact that $T \geq \tilde{T} \implies T^p \geq (\tilde{T})^p \geq \tilde{\sigma} \geq |R(s, a)|$. As a result, $|R(s, a)| \leq 3\mathcal{C}T^p$. Since our analysis above holds for *any* state-action pair, we conclude that $\|\mathcal{T}Q_t\|_\infty \leq 3\mathcal{C}T^p/(1-\gamma)$. With these developments, we can proceed to bound $\zeta_{t,1}$ as follows:

$$
\begin{aligned}
\|\zeta_{t,1}\|_\infty &\leq |\eta_{t,1}(s_t, a_t)| + \|D_t - D\|_\infty (\|Q_t\|_\infty + \|\mathcal{T}Q_t\|_\infty) \\
&\overset{(a)}{\leq} \frac{6\mathcal{C}T^p}{1-\gamma} + (\|Q_t\|_\infty + \|\mathcal{T}Q_t\|_\infty) \\
&\overset{(b)}{\leq} \frac{12\mathcal{C}T^p}{1-\gamma},
\end{aligned}
$$

where (a) follows from (90) and (b) from (89) and the bound we derived on $\|\mathcal{T}Q_t\|_\infty$. This concludes the proof. $\square$

At this stage, it is instructive to compare the bound on $\|\zeta_{t,1}\|_\infty$ from Lemma 4 with that in Lemma 9 above. While in the former, this bound is on the order of $\mathcal{O}(1)$, it is on the order of $\mathcal{O}(T^p)$ in the latter. As a result, if one were to directly use the bound from Lemma 9 in the standard Azuma Hoeffding inequality (much like what we do in Lemma 5), the resulting final bounds would be vacuous. This calls for a more intricate analysis. In this context, our next result provides a finer bound on $\|\zeta_{t,1}\|_\infty$; however, the price of this finer bound is that it now only holds with high probability.

**Lemma 10.** *(Finer Probabilistic Bounds on Iterates for* Robust Async-RAQ*) The following bounds hold with probability at least $1 - 2\delta_1 T$ for all $t \in [T]$:*

$$|\eta_{t,1}(s_t, a_t)| \leq \frac{6\mathcal{C}\tilde{\sigma}}{1-\gamma}, \quad \|\zeta_{t,1}\|_\infty \leq \frac{12\mathcal{C}\tilde{\sigma}}{1-\gamma}, \tag{92}$$

*where $\mathcal{C}$ is the universal constant that appears in (4).*

*Proof.* Let us start by revisiting the bounds on the reward proxy $\tilde{r}_t(s_t, a_t)$ established in Lemma 8. In the proof of Lemma 8, we established that for $t \leq \tilde{T}$, $|\tilde{r}_t(s_t, a_t)| \leq 3\mathcal{C}\tilde{\sigma}$ *deterministically*. Furthermore, we also showed that for $t > \tilde{T}$, the following are true with probability at least

$1 - 2\delta_1 T$: (i) $\tilde{r}_t(s_t, a_t) = \bar{r}_t(s_t, a_t)$, and (ii) $|\bar{r}_t(s_t, a_t)| \leq G_t$, where $G_t$ is as in (4). Since $G_t \leq 3\mathcal{C}\tilde{\sigma}, \forall t \geq T$, we conclude that there exists an event of measure at least $1 - 2\delta_1 T$, on which, $|\tilde{r}_t(s_t, a_t)| \leq 3\mathcal{C}\tilde{\sigma}, \forall t \geq 0$. Restricted to this good event, one can now perform the exact same analysis as in the proof of Lemma 4 to establish the claim of this lemma. □

Based on the previous two results, we now have a martingale difference which exhibits a crude deterministic upper bound, and a finer bound that holds with a fixed high probability. We are in need of a refined version of the Azuma Hoeffding inequality that can exploit this structure. Thankfully, [22, Theorem 7] provides us with precisely the right tool. Our next result is a slight adaptation of this theorem; we provide its proof for completeness.

**Theorem 7.** *Probabilistic Azuma-Hoeffding Inequality [22] Let $X_0, \ldots, X_n$ be a martingale with $X_0$ constant, such that:*

    *(i) With probability $\geq 1 - r$, $|X_{i+1} - X_i| \leq c_i$ for $0 \leq i < n$.*

    *(ii) $|X_{i+1} - X_i| \leq b_i$, deterministically.*

*Assume $b_i \cdot r^{\frac{1}{2}} \leq c_i$. Then, the following holds:*

$$\mathbb{P}\left[|X_n - X_0| > \sqrt{\left(32 \sum_{i=1}^n c_i^2\right) \log\left(\frac{2}{\delta}\right)} + \sum_{i=0}^{n-1} b_i \cdot r^{1/2}\right] < \delta + 2nr^{1/2}. \tag{93}$$

*Proof.* Let $\mathcal{F}_i$ denote the event $|X_{i+1} - X_i| > c_i$. Define a new martingale $\{Y_0, Y_1, \ldots, Y_n\}$ where $Y_0 = X_0$. Additionally, let $p = \mathbb{P}(\mathcal{F}_i|X_i)$. We consider two cases:

    (A) If $p \geq r^{\frac{1}{2}}$, terminate the martingale by setting $Y_j = Y_i$ for all $j \in [i, n]$.

    (B) If $p < r^{\frac{1}{2}}$, and the martingale has not been previously terminated, define:

$$\bar{X}_{i+1} = \begin{cases} X_i & \text{if } \mathcal{F}_i, \\ X_{i+1} & \text{otherwise.} \end{cases}$$

Then,
$$\mathbb{E}[\bar{X}_{i+1}|X_i] = \mathbb{E}[X_{i+1}|X_i] + \mathbb{E}[\bar{X}_{i+1} - X_{i+1}|X_i].$$
Define $A_i \triangleq \mathbb{E}[\bar{X}_{i+1} - X_{i+1}|X_i]$. Then:
$$A_i = \mathbb{E}[\bar{X}_{i+1} - X_{i+1}|X_i, \mathcal{F}_i] \cdot \mathbb{P}(\mathcal{F}_i|X_i).$$

Using the crude bound $|X_{i+1} - X_i| \leq b_i$ and $p = \mathbb{P}(\mathcal{F}_i|X_i) < r^{\frac{1}{2}}$, we obtain:

$$A_i \leq b_i \cdot r^{\frac{1}{2}}.$$

Define the new martingale $\{Y_n\}_{n \geq 1}$, $Y_0 = X_0$ as:

$$Y_{i+1} = Y_i + (\bar{X}_{i+1} - X_i - A_i),$$

which satisfies:

$$|Y_{i+1} - Y_i| \leq c_i + b_i \cdot r^{\frac{1}{2}} \leq 2c_i.$$

Since $\mathbb{E}[Y_{i+1} - Y_i|Y_i] = 0$, $\{Y_n\}_{n \geq 1}$ is a martingale. For the event $\{\mathcal{G}\}$ (where neither case (A) nor $\mathcal{F}_i$ occurs), it follows from the construction:

$$Y_n = X_n - \sum_{i=0}^{n-1} A_i.$$

Therefore, we get the following deterministic bound for the event $\{\mathcal{G}\}$:

$$|Y_n - X_n| = \left|\sum_{i=0}^{n-1} A_i\right| \leq r^{\frac{1}{2}} \sum_{i=0}^{n-1} b_i.$$

For the complementary event $\{\sim \mathcal{G}\}$, we have:

$$
\mathbb{P}(\sim \mathcal{G}) = \mathbb{P}\left(\{\cup_i \mathcal{F}_i\} \bigcup \{\cup_i \mathbb{P}(\mathcal{F}_i | X_i) > r^{\frac{1}{2}}\}\right)
$$

$$
\leq \mathbb{P}(\cup_i \mathcal{F}_i) + \mathbb{P}(\cup_i \mathbb{P}(\mathcal{F}_i | X_i) > r^{\frac{1}{2}})
$$

$$
\overset{(\bullet)}{\leq} \sum_{i=1}^n \mathbb{P}(\mathcal{F}_i) + \sum_{i=1}^n \mathbb{P}[\mathbb{P}(\mathcal{F}_i | X_i) > r^{\frac{1}{2}}] \leq nr + nr^{\frac{1}{2}} \tag{94}
$$

$$
\leq 2nr^{\frac{1}{2}}.
$$

To bound the complementary event $\{\sim \mathcal{G}\}$, we start by expressing it as $\{\cup_i \mathcal{F}_i\} \bigcup \{\cup_i \mathbb{P}(\mathcal{F}_i | X_i) > r^{1/2}\}$. In $(\bullet)$, we used the following inequality $\mathbb{P}(\cup_i A_i) \leq \cup_i \mathbb{P}(A_i)$. Combining these, we obtain $\mathbb{P}(\sim \mathcal{G}) \leq nr + nr^{1/2}$, which simplifies to $\mathbb{P}(\sim \mathcal{G}) \leq 2nr^{1/2}$ since $r \leq r^{1/2}$ for $r \leq 1$. Now, we can finally arrive at the following bound:

$$
\mathbb{P}\left[|X_n - X_0| > \sqrt{\left(32 \sum_{i=1}^n c_i^2\right) \log\left(\frac{2}{\delta}\right)} + \sum_{i=0}^{n-1} b_i \cdot r^{1/2}\right]
$$

$$
\overset{(*)}{\leq} \mathbb{P}\left[|X_n - Y_n| + |Y_n - Y_0| > \sqrt{\left(32 \sum_{i=1}^n c_i^2\right) \log\left(\frac{2}{\delta}\right)} + \sum_{i=0}^{n-1} b_i \cdot r^{1/2}\right]
$$

$$
\overset{(**)}{\leq} \mathbb{P}\left[\left\{|X_n - Y_n| > \sum_{i=0}^{n-1} b_i \cdot r^{1/2}\right\} \cup \left\{|Y_n - Y_0| > \sqrt{\left(32 \sum_{i=1}^n c_i^2\right) \log\left(\frac{2}{\delta}\right)}\right\}\right] \tag{95}
$$

$$
\overset{(***)}{\leq} \mathbb{P}(\sim \mathcal{G}) + \mathbb{P}\left[|Y_n - Y_0| > \sqrt{\left(32 \sum_{i=1}^n c_i^2\right) \log\left(\frac{2}{\delta}\right)}\right]
$$

$$
\leq 2nr^{\frac{1}{2}} + \delta.
$$

In step $(*)$, we apply the triangle inequality, which states that $|X_n - X_0| \leq |X_n - Y_n| + |Y_n - Y_0|$, allowing us to bound the original probability by replacing $|X_n - X_0|$ with $|X_n - Y_n| + |Y_n - Y_0|$. In step $(**)$, we use the union bound, which ensures that $\mathbb{P}(\mathcal{A} + \mathcal{B} > \mathcal{Q}) \leq \mathbb{P}(\mathcal{A} > \mathcal{Q}_1) + \mathbb{P}(\mathcal{B} > \mathcal{Q}_2)$, where $\mathcal{Q}_1 + \mathcal{Q}_2 = \mathcal{Q}$. Finally, in step $(***)$, we use the bound $\mathbb{P}(\sim \mathcal{G}) \leq 2nr^{1/2}$ for the first term, as $|X_n - Y_n|$ is controlled by the good event $\mathcal{G}$, and the second term is bounded by $\delta$ due to properties of $Y_n$. With this, our proof is complete. $\qquad\square$

Armed with the previous result, we are now in a position to control the noise term in `Robust Async-RAQ`.

**Lemma 11.** (***Bounding Non-Adversarial Noise in*** `Robust Async-RAQ`) *Suppose* $\delta_1 \leq \delta^2 / 128 |\mathcal{S}|^2 |\mathcal{A}|^2 T^{2p+3}$. *Then, with probability at least* $1 - \delta/2$, *the following bound holds simultaneously for all* $t \in [T]$:

$$
\left\|\sum_{k=0}^t \alpha(I - \alpha D)^{t-k} \zeta_{k,1}\right\|_\infty \leq \mathcal{O}\left(\frac{\tilde{\sigma}}{1-\gamma}\right) \cdot \sqrt{\frac{\alpha}{\lambda_{\min}} \log\left(\frac{8|\mathcal{S}||\mathcal{A}|T}{\delta}\right)} + \mathcal{O}\left(\frac{\alpha}{1-\gamma}\right), \tag{96}
$$

*where* $\zeta_{k,1}$ *is defined in Eq.* (32).

*Proof.* We now return to bounding the non-adversarial noise term in `Robust Async-RAQ` using the probabilistic variant of the Azuma–Hoeffding inequality outlined in Theorem 7. In our setting, we define the following quantities to be directly substituted into Eq. (93) of Theorem 7:

$$
c_i = \frac{12\mathcal{C}\tilde{\sigma}}{1-\gamma} \cdot \alpha(1-\alpha)^{t-i}, \quad b_i = \frac{12\mathcal{C}T^p}{1-\gamma} \cdot \alpha(1-\alpha)^{t-i}, \quad r = 2\delta_1 T. \tag{97}
$$

To satisfy the condition $b_i \cdot r^{1/2} \leq c_i$ that is required to apply Theorem 7, it suffices to ensure:

$$
(2\delta_1 T)^{1/2} \cdot T^p \leq \tilde{\sigma}. \tag{98}
$$

Since $\tilde{\sigma} \geq 1$, the above condition can be ensured by requiring

$$(2\delta_1 T)^{1/2} \cdot T^p \leq 1, \tag{99}$$

which ensures the applicability of the refined concentration bound. Now, Eq. (99) imposes the following condition on the failure probability: $\delta_1 \leq 1/(2T^{2p+1})$. Assuming that requirement in Eq. (99) holds, then for a fixed $(s, a) \in \mathcal{S} \times \mathcal{A}$ and $t \in [T]$, Theorem 7, when applied with the parameter choices in Eq. (97), implies that with probability at least $1 - \bar{\delta} - 2T(2\delta_1 T)^{1/2}$, the following holds:

$$\left| \sum_{k=0}^{t} \alpha(1 - \alpha\lambda(s,a))^{t-k} \zeta_{k,1}(s,a) \right| \leq \mathcal{O}\left(\frac{\tilde{\sigma}}{1-\gamma}\right) \cdot \sqrt{\frac{\alpha}{\lambda_{\min}} \log\left(\frac{2}{\bar{\delta}}\right)} + \mathcal{O}\left(\frac{\alpha T^{p+1}}{1-\gamma} \cdot (2\delta_1 T)^{1/2}\right). \tag{100}$$

As an immediate next step, applying an union bound over all $(s, a) \in \mathcal{S} \times \mathcal{A}$ and $t \in [T]$, the bound in Eq. (100) holds simultaneously for all state-action pairs and time steps with probability at least

$$1 - \underbrace{|\mathcal{S}||\mathcal{A}|T\bar{\delta}}_{(\bullet)} - \underbrace{2|\mathcal{S}||\mathcal{A}|T^2(2\delta_1 T)^{1/2}}_{(\bullet\bullet)}. \tag{101}$$

Next, we impose the following additional conditions on the failure probability $\delta_1$ to control the second term in Eq. (100), and to ensure that Eq. (100) holds with probability at least $1 - \delta/2$:

$$(2\delta_1 T)^{1/2} \cdot T^{p+1} \leq 1, \quad \underbrace{2|\mathcal{S}||\mathcal{A}|T^2(2\delta_1 T)^{1/2}}_{(\bullet\bullet)} \leq \delta/4. \tag{102}$$

Combining all the constraints on $\delta_1$ from Eq. (99) and Eq. (102), we arrive at the final condition on the failure probability $\delta_1$ as follows:

$$(2\delta_1 T)^{\frac{1}{2}} \leq \delta/(8|\mathcal{S}||\mathcal{A}|T^{p+1}) \implies \delta_1 \leq \delta/(128|\mathcal{S}|^2|\mathcal{A}|^2 T^{2p+3}). \tag{103}$$

Now by ensuring that term $(\bullet) \leq \delta/4$ and applying the final requirement on the failure probability from Eq. (103), we conclude that the following bound holds for all state-action pairs $(s, a) \in \mathcal{S} \times \mathcal{A}$, and $t \in [T]$ with probability at least $1 - \delta/2$:

$$\left| \sum_{k=0}^{t} \alpha(1 - \alpha\lambda(s,a))^{t-k} \zeta_{k,1}(s,a) \right| \leq \mathcal{O}\left(\frac{\tilde{\sigma}}{1-\gamma}\right) \cdot \sqrt{\frac{\alpha}{\lambda_{\min}} \log\left(\frac{8|\mathcal{S}||\mathcal{A}|T}{\delta}\right)} + \mathcal{O}\left(\frac{\alpha}{1-\gamma}\right). \tag{104}$$

Hence, given $\delta_1 \leq \delta/(128|\mathcal{S}|^2|\mathcal{A}|^2 T^{2p+3})$, the following also holds with probability at least $1 - \frac{\delta}{2}$:

$$\left\| \sum_{k=0}^{t} \alpha(I - \alpha D)^{t-k} \zeta_{k,1} \right\|_\infty = \max_{(s,a)\in\mathcal{S}\times\mathcal{A}} \left| \sum_{k=0}^{t} \alpha(1 - \alpha\lambda(s,a))^{t-k} \zeta_{k,1}(s,a) \right|$$

$$\leq \mathcal{O}\left(\frac{\tilde{\sigma}}{1-\gamma}\right) \cdot \sqrt{\frac{\alpha}{\lambda_{\min}} \log\left(\frac{8|\mathcal{S}||\mathcal{A}|T}{\delta}\right)} + \mathcal{O}\left(\frac{\alpha}{1-\gamma}\right) \triangleq \tilde{\Delta}_{t,1}. \tag{105}$$

$\square$

**Finite-Time Rates for** `Robust Async-RAQ` **(Proof of Theorem 4).** Having established bounds on the non-adversarial and adversarial terms via Lemma 11 and Lemma 8, respectively, we proceed by adopting the exact same argument strategy as in Section C for the proof of Theorem 2. Keeping the notation same, in `Robust Async-RAQ`, we define the total perturbation term as $\Delta = \tilde{\Delta}_{t,1} + \tilde{\Delta}_{t,2}$, and mimic the inductive proof of Theorem 2 to establish that the exact same bound as in (63) holds with probability at least $1 - \delta$. Finally, substituting $\alpha = \frac{\log T}{\lambda_{\min} T(1-\gamma)}$, and simplifying, we arrive at the following bound with probability at least $1 - \delta$:

$$\|d_T\|_\infty \leq \frac{\|d_0\|_\infty}{T} + O\left(\frac{\tilde{\sigma}^{1+1/2p}}{(1-\gamma)^{\frac{5}{2}} \lambda_{\min}^{\frac{3}{2}} \sqrt{T}} \log T \sqrt{\log\left(\frac{|\mathcal{S}||\mathcal{A}|T}{\delta}\right)} + \frac{\tilde{\sigma}\sqrt{\varepsilon}}{\lambda_{\min}(1-\gamma)}\right). \tag{106}$$

With this, we complete the proof of the finite-time convergence rate for `Robust Async-RAQ`.

# F   Extension to the Markov Setting and Proof of Theorem 5

---

**Algorithm 3** Robust Asynchronous Q-Learning Algorithm (`Robust Async-RAQ-M`)

---

1: **Input:** Step-size $\alpha$, corruption fraction $\varepsilon$, confidence level $\delta$, mixing time $\bar{\tau}$, iteration count $T$.
2: Initialize datasets $\mathcal{D}_0(s, a) = \emptyset$, for all $(s, a) \in \mathcal{S} \times \mathcal{A}$, and Q-table $Q_0 = 0$.
3: Set block size $\tau = \lfloor \lceil \log(2T/\delta)/ \log 2 \rceil \cdot \bar{\tau} \rfloor$
4: **for** iteration $t = 0, \ldots, T - 1$ **do**
5:     Observe data tuple $\{s_t, a_t, s_{t+1}\}$, and reward $y_t(s_t, a_t)$.
6:     **if** $t \bmod \tau = 0$ **then**             ▷ Update on every `τ-th subsample`
7:         Append $y_t(s_t, a_t)$ to $\mathcal{D}_t(s_t, a_t)$, and compute $\bar{r}_t(s_t, a_t) \leftarrow \mathtt{TRIM}[\mathcal{D}_t(s_t, a_t), \varepsilon, \delta_1]$.
8:         **if** $|\bar{r}_t(s_t, a_t)| > \tilde{G}_t$ in Eq. (7) **then**
9:             Set $\tilde{r}_t(s_t, a_t) \leftarrow 0$
10:        **else**
11:            Set $\tilde{r}_t(s_t, a_t) \leftarrow \bar{r}_t(s_t, a_t)$
12:        **end if**
13:        Update $Q_{t+1}$ using Eq. (5).
14:     **else**
15:        Continue                            ▷ Go to `Line 4`.
16:     **end if**
17: **end for**

---

The goal of this section is to extend our analysis of `Robust Async-RAQ` from the asynchronous i.i.d. sampling setting to the Markov data setting. To keep the paper self-contained, we first present the essential background on the Markovian setting, drawing primarily on [23].

● **Background.** Let $\{Z_t\}$ be an ergodic time-homogeneous Markov chain over a finite-state space $\Omega$ with stationary distribution $\rho$. Define

$$d_{mix}(t) := \sup_{Z \in \Omega} D_{TV}\left(\mathbb{P}(Z_t \in \cdot \mid Z_0 = Z), \rho\right). \tag{107}$$

Then, $d_{mix}(t)$ is a non-increasing function of $t$. We define the *mixing time* as

$$\bar{\tau} := \inf\{t \mid d_{mix}(t) \le 1/4\}. \tag{108}$$

Intuitively, the mixing time measures *how fast the state distribution approaches stationarity*. We then have the following key fact [23]:

$$\boxed{d_{mix}(\ell\bar{\tau}) \le 2^{-\ell}, \quad \forall \ell \in \mathbb{N}.} \tag{109}$$

With the notations specified above, we then introduce the following theorem that will play a crucial role in our extension to the Markov setting.

**Theorem 8** (Coupling). *Let $Z_0, Z_1, \cdots$ be a stationary finite-state Markov chain with stationary distribution $\rho$, and let $K, n \in \mathbb{N}$. Then, we can couple $(Z_0, Z_K, \cdots, Z_{(n-1)K})$ and $(\tilde{Z}_0, \tilde{Z}_K, \cdots, \tilde{Z}_{(n-1)K}) \in \rho^{\otimes n}$, such that*

$$\mathbb{P}\left(\{Z_0, Z_K, \cdots, Z_{(n-1)K}\} \ne \{\tilde{Z}_0, \tilde{Z}_K, \cdots, \tilde{Z}_{(n-1)K}\}\right) \le (n-1)d_{mix}(K). \tag{110}$$

The proof of this theorem can be found in [24]. Intuitively, Theorem 8 states that if we subsample a sequence from an ergodic Markov chain with sufficiently large sampling interval, then with high probability, the sub-sampled sequence is identical to its i.i.d. counterpart sampled from the stationary distribution of that Markov chain. Let us now see how these ideas can be exploited for our setting.

**Extension to the Markov Setting.** Recall that $\mu$ is the behavior policy that generates data in our problem. Let the trajectory generated by this policy be $\{s_0, a_0, s_1, a_1, \cdots\}$. Note that $\{Z_t\} := \{(s_t, a_t, s_{t+1})\}$ is also a Markov chain, and that it is ergodic in light of Assumption 1; see [52]. Suppose this chain is initialized from its stationary distribution $\rho$. Let $\bar{\tau}$ be the mixing time of this Markov chain.

We now propose a simple modification to `Robust Async-RAQ` that is based on dropping certain data points. To see how this can be done, we define a "block" parameter $\tau := \lfloor \ell \bar{\tau} \rfloor$, where $\ell = \lceil \log(2T/\delta)/\log 2 \rceil$. The only modification to `Robust Async-RAQ` is that the agent now uses every $\tau$-th sample, and drops the rest; this variant is formally described in Algorithm 3.

To analyze Algorithm 3, we note that it essentially runs on $n = T/\tau$ samples; for simplicity, we assume that $n$ is an integer. Specifically, the learner only uses the data set $\{Z_0, Z_\tau, \cdots, Z_{(n-1)\tau}\}$. Let $\{\tilde{Z}_0, \tilde{Z}_\tau, \cdots, \tilde{Z}_{(n-1)\tau}\} \sim \rho^{\otimes n}$ be i.i.d. samples drawn from the stationary distribution $\rho$. From the coupling theorem, namely Theorem 8, given any $\delta \in (0,1)$, we then have

$$
\begin{aligned}
\mathbb{P}\left(\{Z_0, Z_\tau, \cdots, Z_{(n-1)\tau}\} \neq \{\tilde{Z}_0, \tilde{Z}_\tau, \cdots, \tilde{Z}_{(n-1)\tau}\}\right) &\leq n d_{mix}(\tau) \\
&\leq n d_{mix}(\lfloor \ell\bar{\tau} \rfloor) \\
&\leq \frac{T}{\tau} \cdot 2^{-\ell} \\
&\leq T \cdot 2^{-\ell} \\
&\leq T \cdot \frac{\delta}{2T} \\
&= \frac{\delta}{2},
\end{aligned}
\tag{111}
$$

where we used the key fact (109), the definition of $\ell$, and the fact that $d_{mix}(t)$ is non-increasing in $t$.

Thus, there exists a "good event", say $\mathcal{B}$, of measure at least $1 - \delta/2$, on which

$$
\{Z_0, Z_\tau, \cdots, Z_{(n-1)\tau}\} = \{\tilde{Z}_0, \tilde{Z}_\tau, \cdots, \tilde{Z}_{(n-1)\tau}\}.
\tag{112}
$$

Equation (112) states that on the good event $\mathcal{B}$, the sub-sampled Markovian data is identical to its i.i.d. counterpart. To see how this result can be exploited, let us recall the guarantee from Theorem 4 when `Robust Async-RAQ` is run on $n = (T/\tau)$ i.i.d. samples with

$$
T > \max\{\tau\bar{T}, \tau \log(T)/(\lambda_{\min}(1-\gamma))\} \quad \text{and} \quad \alpha = \frac{\tau \log T}{\lambda_{\min}(1-\gamma)T}.
$$

In this setting, the following holds with probability $1 - \delta/2$:

$$
\|d_n\|_\infty \leq \frac{\|d_0\|_\infty}{T} + \underbrace{c_1\left(\frac{\tilde{\sigma}^{1+1/2p}}{(1-\gamma)^{\frac{5}{2}}} \frac{\log T}{\lambda_{\min}^{\frac{3}{2}}\sqrt{T}} \sqrt{\tau \log\left(\frac{|\mathcal{S}||\mathcal{A}|T}{\delta}\right)}\right) + c_2\left(\frac{\tilde{\sigma}\sqrt{\varepsilon}}{\lambda_{\min}(1-\gamma)}\right)}_{\Psi},
\tag{113}
$$

where $c_1$ and $c_2$ are suitable universal constants.

Now consider running Algorithm 3 on the $n$ Markov samples $\{Z_0, Z_\tau, \cdots, Z_{(n-1)\tau}\}$, with $Z_0 \sim \rho$. Let the output of the algorithm be $Q_n$ in this case. Keeping everything else the same, let the output of Algorithm 3 be $\tilde{Q}_n$ when it is run on the i.i.d. dataset $\{\tilde{Z}_0, \tilde{Z}_\tau, \cdots, \tilde{Z}_{(n-1)\tau}\}$. From our definition of the event $\mathcal{B}$, we clearly have $Q_n = \tilde{Q}_n$ on event $\mathcal{B}$. Based on this, observe

$$
\begin{aligned}
\mathbb{P}(\{\|Q_n - Q^*\|_\infty > \Psi\}) &= \mathbb{P}(\{\|Q_n - Q^*\|_\infty > \Psi\} \cap \mathcal{B}) + \mathbb{P}(\{\|Q_n - Q^*\|_\infty > \Psi\} \cap \mathcal{B}^c) \\
&\leq \mathbb{P}(\{\|Q_n - Q^*\|_\infty > \Psi\} \cap \mathcal{B}) + \mathbb{P}(\mathcal{B}^c) \\
&\overset{(a)}{\leq} \mathbb{P}(\{\|\tilde{Q}_n - Q^*\|_\infty > \Psi\} \cap \mathcal{B}) + \mathbb{P}(\mathcal{B}^c) \\
&\overset{(b)}{\leq} \mathbb{P}(\{\|\tilde{Q}_n - Q^*\|_\infty > \Psi\}) + \delta/2 \\
&\overset{(c)}{\leq} \delta.
\end{aligned}
\tag{114}
$$

In the above steps, for (a), we used the fact that $Q_n = \tilde{Q}_n$ on event $\mathcal{B}$. For (b), we appealed to (111), and for (c), we used (113). Thus, via the coupling argument above, we have established that with probability at least $1 - \delta$, the following is true:

$$
\|Q_n - Q^*\|_\infty \leq \Psi,
$$

with $\Psi$ as in (113). This is precisely what was needed to be shown to establish Theorem 5.

