# OpenReview forum: "Corruption-Tolerant Asynchronous Q-Learning with Near-Optimal Rates"
_NeurIPS.cc/2025/Workshop/Reliable_ML — NeurIPS 2025 - Reliable ML Workshop_

### Official Review · Reviewer_ZULz · 2025-09-17
**This paper is out of my research area and I can not provide opinion**

**Rating:** 6
**Confidence:** 2

**Review:**

## Summary

This paper proposes a novel robust variant of the Q-learning algorithm and analyzes it under the challenging asynchronous sampling model with time-correlated data

The work is out of my research area, and I can not provide an opinion on this paper.

---

### Official Review · Reviewer_VCDp · 2025-09-19
**Review of Submission 84**

**Rating:** 8
**Confidence:** 3

**Review:**

** Summary**

They study adversarial robustness in RL with infinite horizon and online, sequential data. Specifically, they aim to study two questions: 1) can one still reliably estimate the optimal state-action value function? 2) what is a fundamental lower bound on estimation accuracy in this setting?

For q1, they propose a robust asynchronous Q-learning algorithm and proves a high-probability finite-time bound that matches the usual rate up to an additive.

For q2, They also prove an information-theoretic lower bound and establishes near-optimality in their setting.
Strengths

The paper is explicit and rigorous in answering both of the questions stated and provides tight lower bounds in its chosen setting. It provides the first formal adversarial-robustness guarantees for async Q-learning. It presents a clear, self-contained theory story with an algorithm, finite-time guarantees, and a matching lower bound.

**Weaknesses**

The exposition is dense and heavily theorem-driven, which can make it hard for readers who are not experts to see the broader intuition. It also lacks motivating examples or illustrations to show why corruption matters in practice, leaving the reader with mainly technical results.

**Suggestions for Authors**

 The paper could benefit from a short motivating scenario or even a simple synthetic experiment to help situate the theory in a more concrete narrative. This would make the contributions more accessible and highlight their real-world significance.